

# Precipitation Transition Regions over the Southern Canadian Cordillera during January–April 2010 and under a Pseudo-Global Warming Assumption

Juris D. Almonte[a] and Ronald E. Stewart

Department of Environment and Geography, University of Manitoba, Winnipeg, Manitoba, Canada, R3T 2N2
[a] now at: Environmental Science and Engineering Program, University of Northern British Columbia, Prince
George, British Columbia, Canada, V2N 4Z9
*Correspondence to:* Juris D. Almonte (juris.almonte@unbc.ca)

**Abstract.** The occurrence of various types of winter precipitation is an important issue over the southern Canadian
Cordillera. This issue is examined from January to April of 2010 by exploiting the high-resolution Weather
Research and Forecasting (WRF) model Version 3.4.1 dataset that was used to simulate both a historical reanalysis
(CTRL) and a Pseudo-Global Warming (PGW) experiment (Liu et al., 2016). Transition regions, consisting of both
liquid and solid precipitation or liquid precipitation below 0° C, occurred on 93 % and 94 % of the days in the
present and PGW future, respectively. This led to accumulated precipitation within the transition region increasing
by 27 % and was associated with a rise in its average elevation by 374 m over the Coast and Insular Mountains and
by 240 m over the Rocky Mountains and consequently to an eastward shift towards the higher terrain of the Rocky
Mountains. Transition regions comprised of only rain and snow were most common under both the CTRL and PGW
simulations although all seven transition region categories occurred. Transition region changes would enhance some
of the factors leading to avalanches and would also impact ski resort operations.

## 1 Introduction

The phase of precipitation at the surface is paramount to many processes within the water cycle and it also affects
the way we manage water and at times its risk to society. During the cold season, precipitation can be solid (snow),
liquid (rain, freezing rain) or mixed (wet snow for example). The precipitation-type transition region, where mixed
or freezing precipitation occurs, lies between areas of all rain and all snow if they both occur. Midlatitudes regions,
particularly over many mountains, have frequent transition region occurrences during the cold season, as the 0° C
isotherm can be situated anywhere along the mountainside.

The transition region has significant impacts on the transportation, tourism and water management sectors. For
example, precipitation in the transition region, such as freezing rain, can bring transportation to a halt on major
highways such as happened on the Coquihalla Highway in British Columbia in 2017 (Canadian Press, 2017). The



order of precipitation type occurrence when surface temperatures are near 0° C also has impacts on the snowpack
that can lead to more snow avalanches (COMET, 2010). From a hydrological perspective, the transition region
demarcates the lower temperature threshold of precipitation only falling as rain; this rain can lead to runoff and
eventual flooding (Lundquist et al., 2008).

Transition regions across Canada have been studied for some time (Stewart and King, 1987; Stewart and
Mcfarquhar, 1987;  Stewart, 1992; Stewart et al., 1995; Cortinas et al., 2004; Theriault et al., 2012; Groisman et al.,
2016). Many of these studies focussed on individual events but, in terms of climatology, Cortinas et al. (2004)
carried out an analysis of hazardous winter precipitation types (ice pellets, freezing rain and freezing drizzle) that
can be found within the transition region. They found preferential regions of freezing rain and ice pellets over the
Rocky Mountains using available manual observational data. However, due to the paucity of human observers
located over the Canadian Cordillera and the available information being overwhelmingly within valleys, these
findings may not be representative of what is occurring at higher elevations. A recent study by Groisman et al.
(2016) investigated changes in climatological occurrences of freezing precipitation. They compared a recent period
of (2005–2014) to a 30 year base climatology (1974–2004) and found that the annual number of days with freezing
precipitation increased from 0.1 days to > 3 days over some regions within southern British Columbia.


Accurately simulating the transition region is difficult, especially over orographic regions, where dramatic variations
in terrain increase the complexity of the interacting governing processes (Stoelinga et al., 2003; Minder et al., 2011;
Ikeda et al., 2013; Marks et al., 2013). The elevation of the transition region furthermore moves up and down during
a storm and it varies from storm to storm (Marks et al., 2013). The grid scale resolution and microphysical
parameterizations used are important factors to consider when simulating such transition regions (Ikeda et al., 2013).
A coarse resolution model is not able to capture the orographic processes occurring and therefore tends to
underestimate precipitation amounts as shown, for example, in a sensitivity experiment by Ikeda et al. (2010) using
high-resolution modelling at different horizontal scales from 2 to 36 km.

An opportunity to begin to address transition regions within the southern Canadian Cordillera did not occur until
recently. A major atmospheric field campaign, the Science and Nowcasting of Olympic Weather for Vancouver
2010 (SNOW-V10), was held in conjunction with the 2010 Vancouver Winter Olympic and Paralympic games. This
campaign sought to improve winter weather forecasting within complex terrain, showcasing the difficulty of
forecasting for precipitation types within the transition region (Thériault et al., 2012; Isaac et al., 2014).


The Coast Mountains, next to the Pacific Ocean, often experience enhanced precipitation due to the interactions
between its terrain and the advection of warm moist air (Houze, 2012). During the SNOW-V10 period, the Olympic
venues experienced several issues with warm weather, which led to delayed events over Cypress Mountain, whereas
Whistler, at a higher elevation, received a great deal of snow, but also experienced several transition region
occurrences (Goldenberg, 2010; Guttsman, 2010; Thériault et al., 2012; Isaac et al., 2014). Thériault et al. (2012)
highlighted the dynamical effects of the diabatic cooling from melting, which resulted in the reversal of the valley
flow at the base of Whistler Mountain. Thériault et al. (2012) reviewed five storms occurring over Whistler



Mountain, including two transition region events. Although temperatures were conducive for melting particles, the often subsaturated environment led to the sublimation of much of the falling precipitation (Thériault et al., 2012).

Another study by Berg et al. (2017) pointed out that precipitation particle trajectories sometimes inhibited precipitation from accumulating at the base of Cypress Mountain from strong advection; the associated vertical air motions were greater than the fall speeds of the precipitation particles.

However, to date no comprehensive regional wide study on the changing transition region over the southern

Canadian Cordillera, which includes the Rocky Mountains, has taken place. As global temperatures rise, it is expected that there will be changes to the transition region locations and elevations over this as well as other regions. Moreover, orographic regions are prone to enhanced warming (Mountain Research Initiative EDW Working Group, 2015). It is important to begin to address the transition region future characteristics.

Such regional studies are now feasible because of recent model developments. In particular, the National Center of Atmospheric Research (NCAR) has carried out 4 km resolution simulations using the Weather Research and Forecasting (WRF) model focussed over the contiguous United States but also including southern Canada and northern Mexico (Liu et al., 2016). These simulations focussed on a multi-year control period in the recent past as well as the same period but under warmer and more moist future conditions using a pseudo-global warming

approach. This high-resolution dataset over the contiguous United States (HRCONUS) is discussed in more detail in Sect. 2.

Given the importance of the transition region and its implications to society, the goal of this article is to address this issue. This study's specific objectives are to analyze the transition region during the 2010 SNOW-V10 project using

data from the WRF 4 km simulations with emphasis on the most severe events, and to examine changes to the transition region under an assumed warmer and more moist climate.

This article is organized as follows. The WRF model setup and criteria for a transition region are outlined in Sect. 2. The evaluation of the WRF model using observational datasets is presented in Sect. 3. An overview of the transition

region under the control and pseudo-global warming simulations is discussed in Sect. 4. Changes to the transition region are presented in Sect. 5 and societal implications under a PGW approach are presented in Sect. 6. Concluding remarks are given in Sect. 7.

## 2 Experimental set-up and methodology

### 2.1 WRF dataset

Two 13 year high-resolution convective-permitting simulations from 2000 to 2013 were carried out by Liu et al. (2016) using the Weather and Research Forecasting (WRF) model version 3.4.1. First, a control (CTRL) run of the historical conditions using the ECMWF Re-Analysis (ERA)-interim was carried out followed by a second using a



Pseudo-Global Warming (PGW) assumption first used by Schär et al. (1996). This PGW method allows one to study the thermodynamic effects of a warmer, more moist climate on historical synoptic systems. In particular, this was accomplished by perturbing the CTRL with the Coupled Model Intercomparison Project 5 (CMIP5) (Taylor et al., 2012) ensemble mean high emissions scenario at the end of the 21st century. These data were utilized in this article because of the high-resolution over complex terrain, availability and ease of access. Comprehensive details on the

WRF model setup can be found in Liu et al. (2016).

The simulations covered the contiguous United States and included southern Canada and northern Mexico. Given this large dataset and areal coverage, a subset from January–April 2010 was extracted to include the southern Canadian Cordillera from the foothills of the Rocky Mountains, to parts of Vancouver Island (-115° W to -127° W)

and from 49° N up to 53° N (Fig. 1). This includes the locations of the SNOW-V10 campaign, but also covers many ski destinations in the southern Canadian Cordillera. Moreover, this study area is not adjacent to the boundary of the HRCONUS domain and should therefore not be affected by lateral boundary effects (Liu et al., 2016).

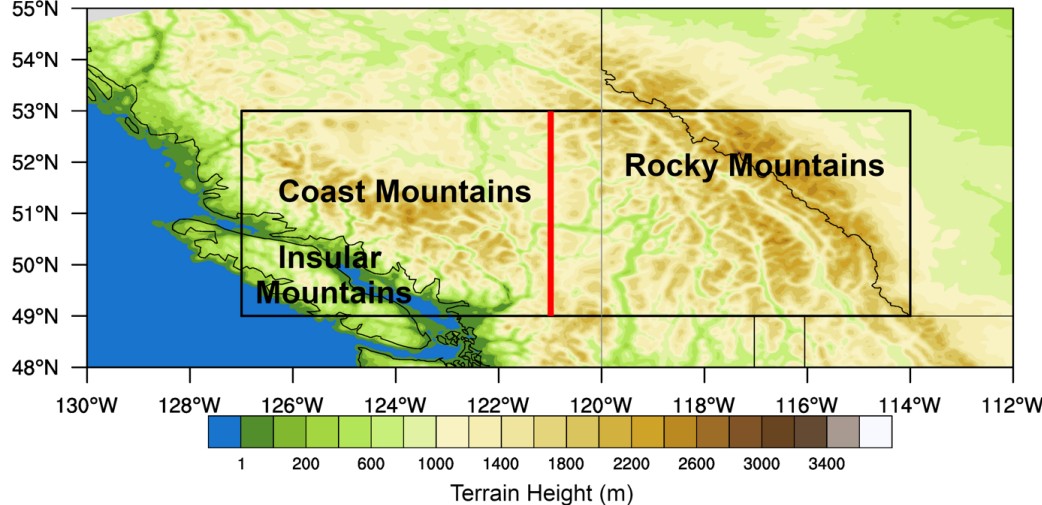

**Figure 1.** Map of study area with terrain height shaded over the Southern Canadian Cordillera, delineated by the
solid black rectangle from 114° W to 127° W and from 49° N to 53° N. The red vertical line at 121° W separates the study area into western and eastern sub-areas.

**2.2 Transition region definition**


Transition regions occurring at the surface are the main focus of this study. These regions are located where mixed precipitation phases occur, including the transition from liquid to solid such as freezing rain and where both solid and liquid-phase precipitation accumulate at the surface simultaneously. Specifically within the model, this refers to





the grid point precipitation accumulations at the lowest level, where a transition region is defined at the surface on
an hourly basis.

It is important to note that 0.2 mm was used as a threshold of each explicit output on an hourly basis. However,
Environment and Climate Change Canada (ECCC) considers an accumulation of 0.2 mm to be a trace amount over a
24 hour period. Through this higher threshold, this study is examining transition regions with substantial
precipitation and it will therefore underestimate the total number of occurrences.

The transition region was broken down into seven categories according to their constituent precipitation (Table 1).
These were categorized in a similar manner to the general criteria of transition regions as discussed above with
additional steps to categorize the type of precipitation. To separate the transition types into their respective
categories, there were checks used to determine when one precipitation type was missing. For example, under a
rain–snow transition category, a criterion to make sure that graupel was less than the 0.2 mm threshold was used.
The rain–snow–graupel category included all precipitation types ≥ 0.2 mm, except for freezing rain, which was
filtered out, using the wet bulb 2 m temperatures > 0° C criterion. The freezing rain–snow–graupel category was
similar, except it used a wet bulb 2 m temperatures ≤ 0° C criterion to exclude rain.

## 3 Evaluation of WRF model

### 3.1 Temperature and relative humidity

Surface temperature and relative humidity are essential for identifying the transition region and diagnosing the type
of precipitation (Matsuo et al., 1981). To evaluate these variables from the CTRL simulation over the January–April
2010 period, ECCC hourly data were retrieved from nine stations (Fig. 2). Table 3 shows average values over this
four month period. Stations were chosen according to their proximity to ski resorts located at higher elevations and
their precipitation availability. Of the nine stations, Glacier National Park (NP) Rogers Pass, Yoho NP and Fernie
had missing data or did not have hourly data and therefore were not included in the comparison. Temperature and
relative humidity were extracted from the closest CTRL grid point to the ECCC station.





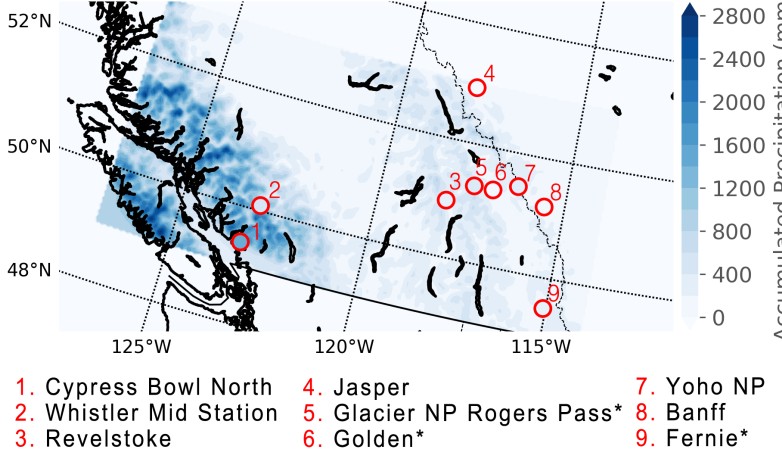

1. Cypress Bowl North    4. Jasper                     7. Yoho NP
2. Whistler Mid Station  5. Glacier NP Rogers Pass*    8. Banff
3. Revelstoke            6. Golden*                    9. Fernie*


**Figure 2**. Map of CTRL total accumulated precipitation (mm) from January to April, 2010 over the study area. Circles indicate the nine ECCC stations used in the precipitation evaluation.

In general, the CTRL was able to simulate the observed temperatures with average values within ±2.5° C at the
stations (Table 2). Overall, there is a tendency to overestimate the temperature at Cypress Bowl North, Whistler Mid Station, Revelstoke and Jasper but underestimate temperature at Banff and Golden. However temperatures at Golden are only reported during manual observation hours during the day; this explains the higher than expected temperatures. Note that 4 km horizontal resolution is still relatively coarse relative to the orographic terrain and in part led to the differences in average temperature between the ECCC stations and the WRF model. For example, the
WRF elevation at Banff was 183 m higher than the actual station. Assuming a saturated adiabatic lapse rate, this would translate to an average temperature of -1.8° C, closer to the Banff station temperature average of -1.5° C. In contrast, at Jasper Warden, although the WRF elevation is higher than the station, the average temperature is still higher under the CTRL.

The CTRL relative humidities were lower (5–27%) when compared to the ECCC stations. The largest discrepancy was at Revelstoke particularly at the beginning of the study period from January to mid-March. For transition regions, these subsaturated surface conditions could mean that the model may be underestimating precipitation at the surface due to sublimational or evaporative losses and it could have an effect on the type of transition since the melting process is slowed.


### 3.2 Precipitation

### 3.2.1 Evaluation using the Canadian Precipitation Analysis



The Canadian Precipitation Analysis (CaPA), a gridded product that uses multiple sources, including radar, observational data, and model data that relies on the Global Multiscale Model (GEM) has been used to evaluate real-time precipitation amounts (Lespinas et al., 2015). The CTRL accumulated precipitation from January to April, 2010 was re-gridded from 4 km to the coarser 10 km CaPA grid, for grid-to-grid comparison, using a conservative spatial interpolation method. The accumulated precipitation for both the CTRL and CaPA, along with their difference is

shown in Fig. 3. Overall, the model has a positive bias over the four month period of 26%, with maximum biases over the Coast Mountains (Fig. 3c).

Spatial correlation of precipitation between the CTRL and CaPA was computed using the Pearson product-moment coefficient of linear regression. There was an overall good agreement in the spatial correlation between the two grids

for the four month period of 0.839. Moreover, the CTRL precipitation was within observational uncertainty over the the western cordillera of the United States, extending up to the Canadian border. As this region borders the Canadian Cordillera and the same geography can be extended across the political boundary, we assume that precipitation amounts simulated under the CTRL are likely within observational uncertainty over the southern Canadian Cordillera as well.


The CaPA dataset comes with its own set of issues that leads to an underestimation of solid precipitation amounts, particularly during the cold season, whereby solid precipitation > 5 mm reports are rejected under routine quality and a heavy reliance on the GEM model for precipitation amounts (Lespinas et al., 2015). However, the GEM model only implicitly accounts for orographic effects with a 15 km resolution (Lespinas et al., 2015). This coarse

resolution is insufficient to resolve complex terrain effects within the Canadian Cordillera and likely underestimates precipitation. Another consideration is the location of the ECCC sites assimilated into CaPA, mostly located within the valleys (Lespinas et al., 2015); these lead to an unproportional representation of the complex terrain.





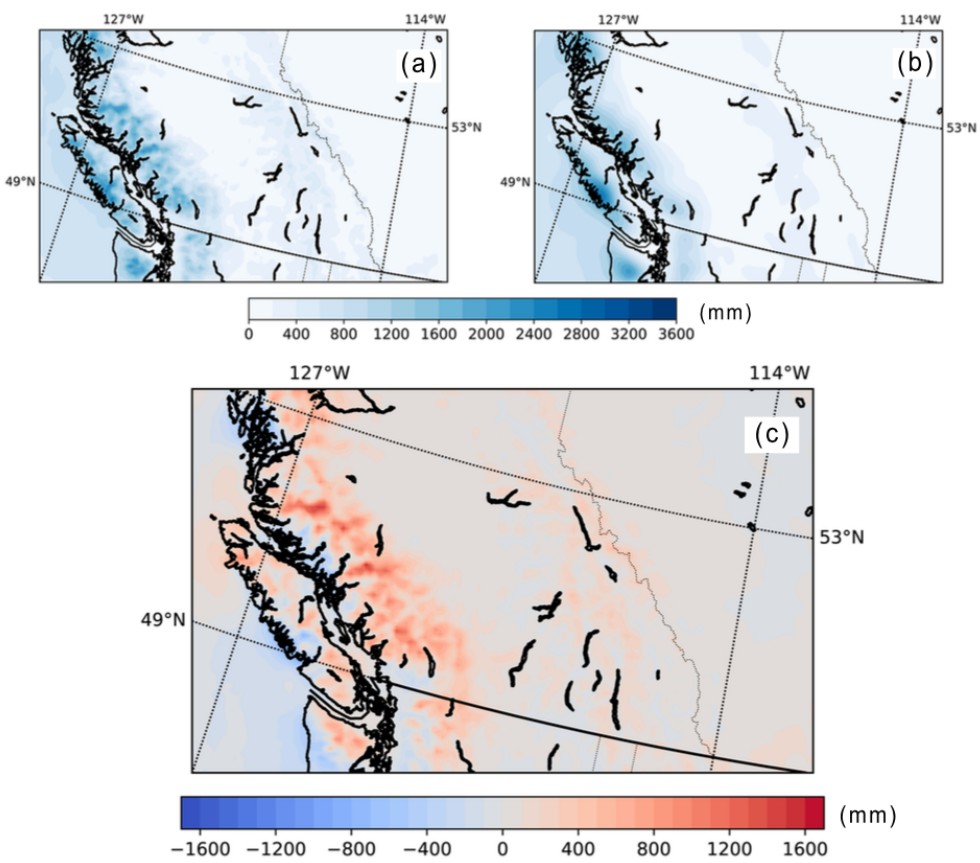


**Figure 3.** (a) Map of CTRL accumulated precipitation (mm) from January to April, 2010. Accumulated precipitation is re-gridded onto a coarser 10x10 km grid, according to CaPA using a conservative interpolation. (b) Map of CaPA accumulated precipitation (mm) from January to April, 2010. (c) Map of accumulated precipitation
difference (CTRL-CaPA) in mm from January to April, 2010.

### 3.2.2 Evaluation using observational stations

ECCC total monthly precipitation amounts are compared to the nearest WRF model grid point. Adjusted data by
Mekis and Vincent (2011) were used where available and these include Glacier NP Rogers Pass, Golden and Fernie.
The variability in accumulated precipitation is displayed, reporting the range of precipitation accumulations from the
eight neighbouring grid points (Fig. 2 and 4).

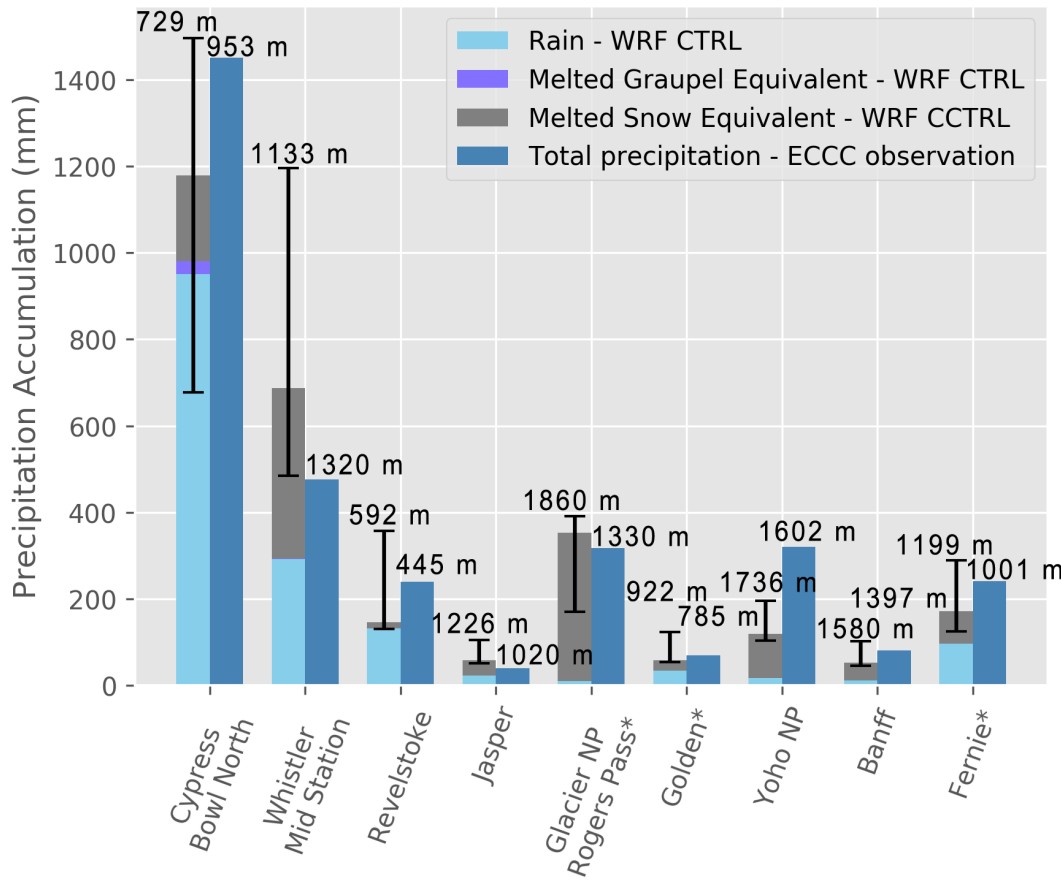

**Figure 4.** Total precipitation accumulation from January to April, 2010. The left-hand bars represent the closest
CTRL grid point, separated into three components, including rain (including freezing rain), snow and graupel. The
vertical bars represent the minimum and maximum total precipitation accumulation of the neighbouring grid points.
The right-hand bars display the ECCC unadjusted station observation. Elevations for the model and ECCC station
are marked above each respective column. * Stations using adjusted precipitation data (Mekis and Vincent 2011).

Over the nine ECCC stations, the CTRL had a positive bias at Whistler Mid Station, Jasper and Glacier NP Rogers
Pass with values of 10–31 %. These stations are located in regions where the CTRL has a positive bias to the CaPA
data. The CTRL had a negative bias at Cypress Bowl North, Revelstoke, Golden, Yoho NP, Banff and Fernie by 19–
168 %, with Yoho NP being the most underestimated station. Albeit, the observed precipitation amounts at the
ECCC stations with a positive bias fell within the precipitation range of the neighbouring grid points, except for
Yoho NP.





Liu et al. (2016) carried out a 13 year evaluation of the CTRL precipitation snow telemetry sites (SNOTEL) across the Western Cordillera of the United States. They used inverse-distance weighted average interpolation of the four closest model grid points to the SNOTEL station and found an overall negative bias of -2 %. They also found a Pearson correlation of 0.9 during the cold season (November-April), suggesting that the CTRL was able to realistically simulate orographic precipitation.

Overall, this evaluation of precipitation over the nine ECCC stations is in good agreement with the retrospective CTRL evaluation conducted over the Western Cordillera of the United States using SNOTEL stations (Liu et al., 2016). A low bias was found over the Canadian Rocky Mountains at Revelstoke, Golden, Yoho NP, Fernie and Banff, similar to the findings of Liu et al. (2016) where a low bias during the cold season over the Colorado Rocky Mountains was found.  In contrast, a high bias of 31 % was found over Whistler, which matches the high bias findings over the Cascade Range (Liu et al., 2016).

Biases found in this study are not unusual as precipitation gauges are subject to undercatchment and may explain some of the CTRL positive bias to CaPA by 26 %.  For example, a recent study by Pan et al. (2016) found that the precipitation gauge underestimated precipitation up to 32.6 % at Marmot Creek, a high elevation station in the Kananaskis area of Alberta.

**4 Occurrence, overall precipitation and locations of transition regions**

Transition regions were very common during the January–April 2010 period. Precipitation occurred on 99 % of the days in both the CTRL and PGW simulations. Of these, 93 % (94 %) had a transition region occurrence. The actual days with a transition region occurrence in the CTRL and PGW simulations were nearly identical (Fig. 5 and 6). This is expected as the storm tracks are constrained by spectral nudging that kept synoptic scale forcings the same. However, the occurrence, spatial distribution and amount of precipitation in transition regions varied greatly from the CTRL to the PGW and are discussed below.



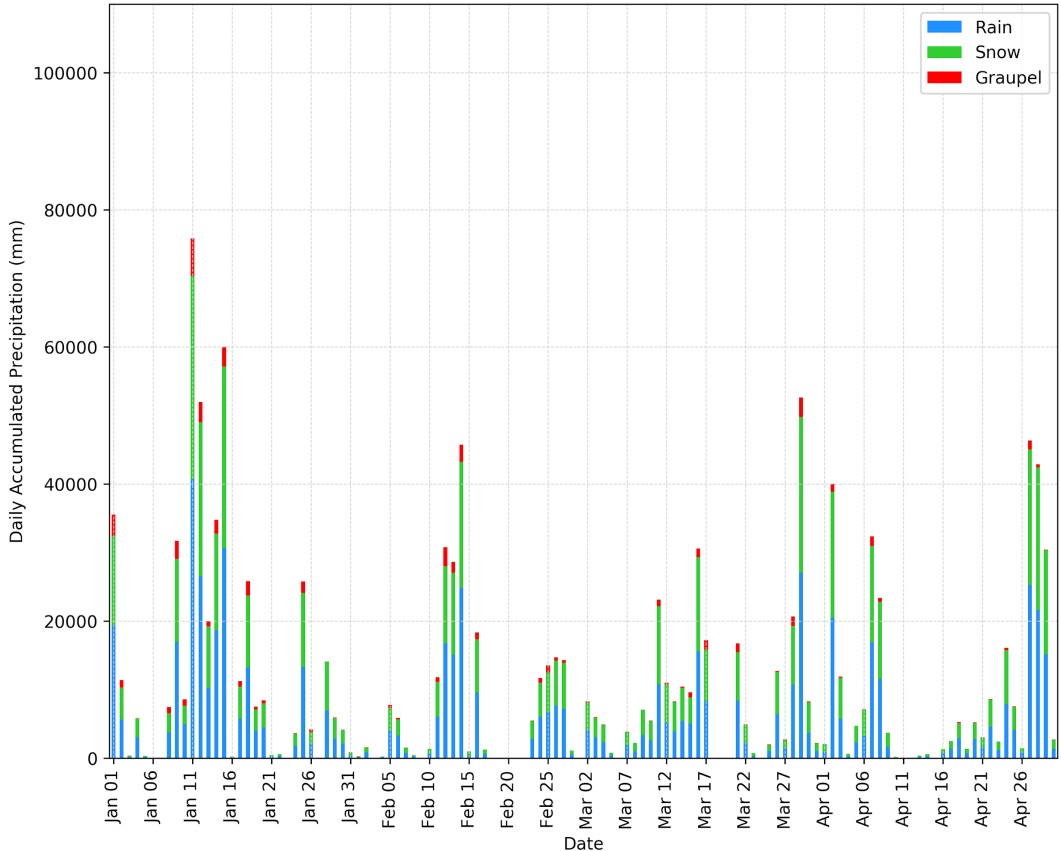

**Figure 5.** Daily accumulated precipitation (mm) within transition regions under CTRL over the January–April, 2010 period and organized by rain (blue), snow (green) and graupel (red). Freezing rain occurrence was minimal and was not shown.






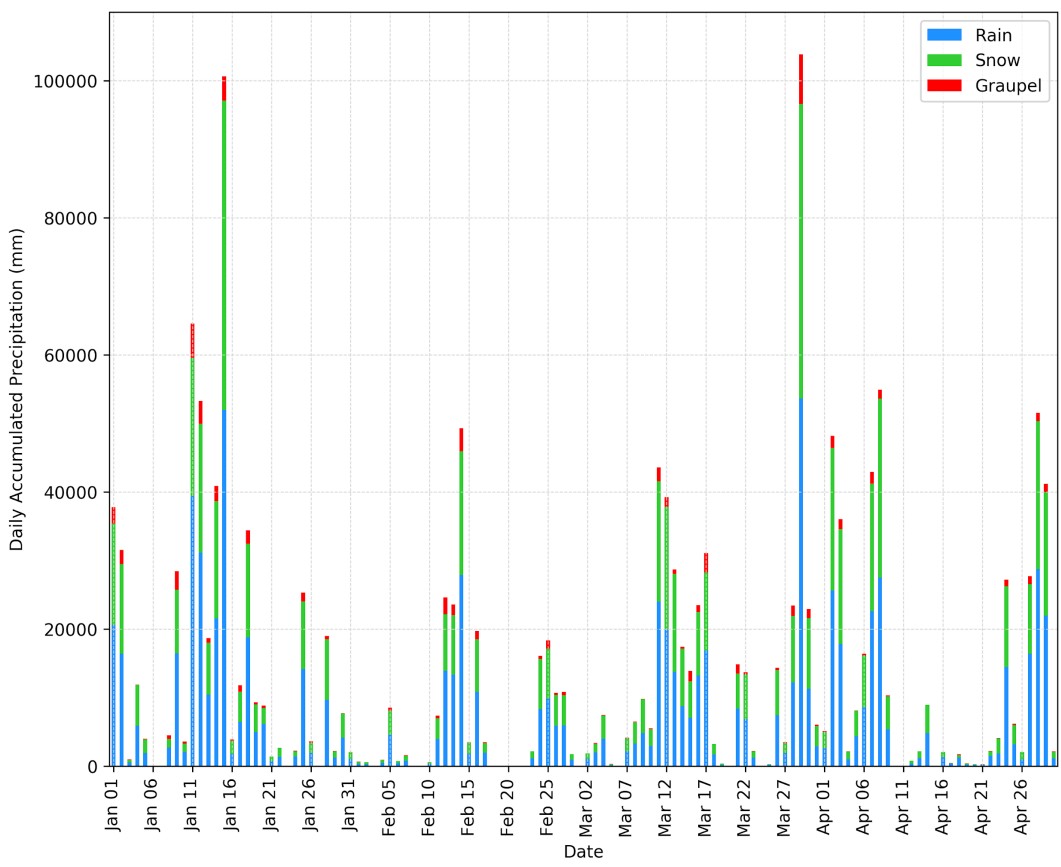


**Figure 6.** Daily accumulated precipitation (mm) within transition regions under PGW over the January–April, 2010 period, organized by rain (blue), snow (green) and graupel (red). Freezing rain occurrence was minimal and was not shown.


The total accumulated precipitation within the transition region increased from 1,263,044 mm (CTRL) to 1,606,163 mm (PGW) a 27% increase and it accounted for ~12% (13%) of the total precipitation under the CTRL (PGW). On average the hourly transition region occupied an area of 4,608 km$^2$ under the CTRL and 5,712 km$^2$ under PGW an increase of 1,104 km$^2$. Given the whole domain's area of 407,696 km$^2$, transition regions represented ~1.1% (1.4%)

of the spatial area under CTRL (PGW). This precipitation mainly occurred over the Insular and Coast Mountains under both the CTRL and PGW. However, under the PGW, less precipitation was simulated over the Insular Mountains and more over the Coast Mountains, especially its leeward side (Fig. 7).






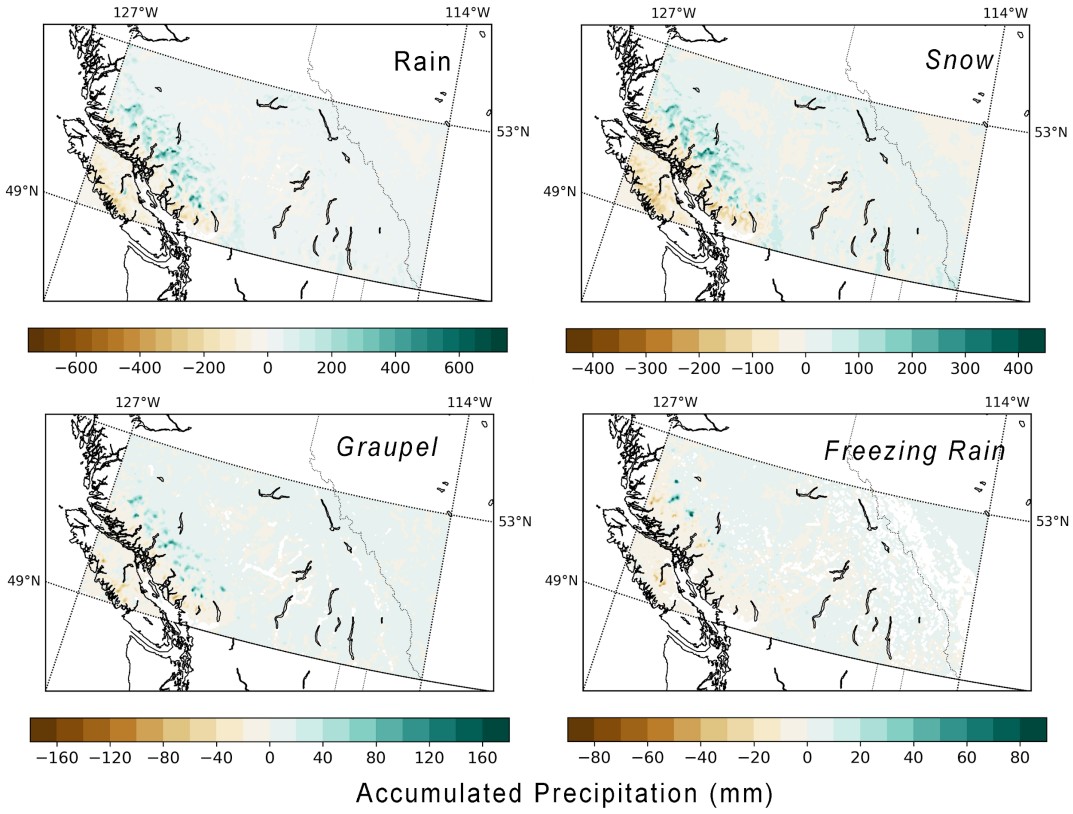

**Figure 7.** Transition region accumulated precipitation difference for snow, graupel, freezing rain and rain (PGW-CTRL) over the January–April, 2010 period.

There was a shift in the distribution of accumulated precipitation types within transition regions from CTRL to PGW (Fig. 8). The proportion of snow decreased from 43.7 % (CTRL) to 41.4 % (PGW), corresponding with an increase in rain from 50.2 % (CTRL) to 52.7 % (PGW). The proportion of graupel remained the same at 4.8 %. There was a small decrease in the proportion of freezing rain from 1.3 % (CTRL) to 1.1 % (PGW).





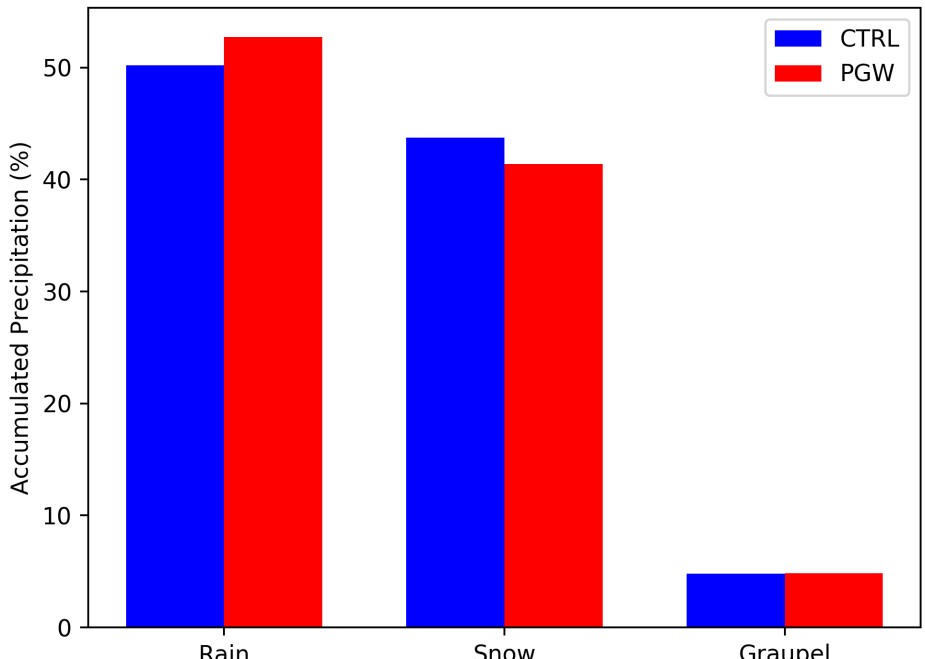

**Figure 8.** Distribution of accumulated snow, rain and graupel within transition regions under the CTRL (blue) and
PGW (red) simulations over the January–April, 2010 period. There was very little freezing rain and it is not shown.

## 5 Characteristic changes to the transition region under historical and future climates

### 5.1 Transition region categories

Over the four month period, all seven transition region categories occurred within the CTRL and PGW simulations
and in both the western and eastern sub-areas (Table 3).

Over the western sub-area, the proportions of the transition region categories did not vary greatly from the CTRL to
PGW, although every change was significant ($\alpha < 0.01$). For example, rain–snow–graupel increased proportionally
from 12.7 % to 14.3 % under PGW, while rain–snow proportionally decreased from 70.1 % to 68.7 %, rain–graupel
proportionally decreased from 3.7 % (CTRL) to 2.0 % (PGW) as did the freezing rain–snow category which
decreased from 5.6 % (CTRL) to 4.3 % (PGW). There were no proportional changes to the freezing rain and
freezing rain–graupel categories.





Similar statistically significant proportions ($\alpha < 0.01$) occurred over the eastern sub-area, but with fewer transition

region occurrences. The rain–snow category also dominated, with a decrease from 84.6 % to 84.0 %. The freezing rain–snow category also decreased from 8.3 % to 5.0 %. The rain–snow–graupel category proportionally increased from 0.6 % to 3.4 %, rain–graupel increased proportionally from to 0.6 %, freezing rain also increased proportionally from 6.1 % to 6.7 %. The rain–snow–graupel, rain–graupel, freezing rain–snow–graupel and freezing rain–graupel categories each comprised of < 1.0 % of the total transition region occurrences.


### 5.2 Vertical and horizontal movement

Various factors affected the vertical movement of transition regions. First, the widespread increase in temperatures

under the PGW simulations naturally led to an upward movement of the transition region.

Second, the vertical movement was accentuated by Elevation Dependent Warming (EDW), whereby higher elevations are more prone to greater warming due to several factors, including the loss in snow cover (Mountain Research Initiative EDW Working Group, 2015). This introduced a non-linear movement that was most apparent

over the regions with greatest snow cover loss (900-1,500 m), due to the surface albedo change (Fig. 9). This enhanced temperature increase of 4.1° C contributed to the elevation increase over both the eastern and western sub-areas.

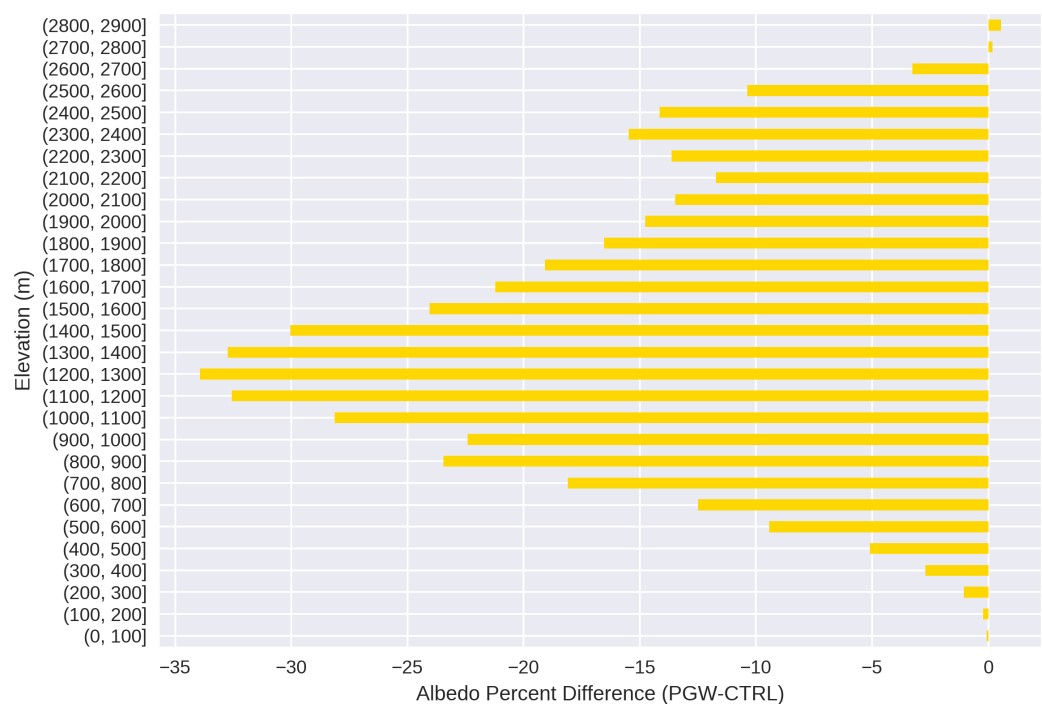





**Figure 9.** Surface albedo percentage difference (PGW-CTRL) within the study area binned at 100 m intervals from
0 - 3,000 m over the January–April, 2010 period.

Third, PGW precipitation rates increased (Sect. 4). The associated diabatic cooling from the melting and/or
evaporation of falling precipitation counters the effects of a rising transition region and this factor would
consequently have lessened the effect of rising temperatures. Precipitation rate is one of the key parameters in
determining the vertical movement of the transition region (Stoelinga et al., 2013).

Specifically, transition regions in the western and eastern sub-areas varied in average elevations (Fig. 10). In the
western sub-area the average transition region elevation under the CTRL was 971 ± 377 m (mean ± sd), and
increased to 1,345 ± 397 m under PGW, an increase of 374 m. Over the eastern sub-area, the average elevation for a
transition region under the CTRL was 1,351 ± 346 m and 1,591 ± 327 m under the PGW, an increase of 240 m (Fig.
10). The transition region on average was lower, but the increase was greater over the Coast Mountains than over the
Rocky Mountains.

There are three explanations for the difference in average elevation. First, the number of transition regions was 3.5
times greater over the western sub-area, often occurring below 1,000 m (55%) and near sea level under CTRL,
whereas the lowest elevation in the eastern sub-area is 333 m (Table 3). Since each sub-area had transition regions
up to approximately the same elevation, the average is lower in the west. Secondly and related, when warm moist
Pacific air entered the study area, the elevation of the 0° C isotherm would at times occur above the peaks of the
Insular and Coast Mountains, effectively lowering the average elevation of the transition region. In contrast, over the
Rocky Mountains, the same Pacific air was the dominant cause of transition regions but the 0° C isotherm still
occurred below orographic peaks. Thirdly, the precipitation rate in transition regions was higher over the Insular and
Coast Mountains, which, as previously mentioned, would lower its elevation due to diabatic cooling.

The greater increase in average transition region elevation over the western sub-area can also be explained (Fig. 10).
Temperature differences at the elevation layer of the average transition region were similar over both sub-areas
(approximately 4.1° C) which would imply an expected increase of 600–700 m assuming a saturated adiabatic lapse
rate. Actual values were approximately 55 % and 35% of this over the western and eastern sub-areas, respectively.
In the western sub-area, snow did not occur often even in CTRL, so PGW warming simply meant that transition
regions moved upwards except for those near peaks which were eliminated. In the eastern sub-area, snow often
occurred in CTRL with no transition regions so PGW warming led to their occurrence but at low elevation; existing
transitions with CTRL simply moved upwards. For the conditions of these particular simulations, the elimination of
high elevation transition regions over the western sub-area reduced its average elevation increase less than the
contribution of additional, low elevation transition regions over the eastern sub-area.






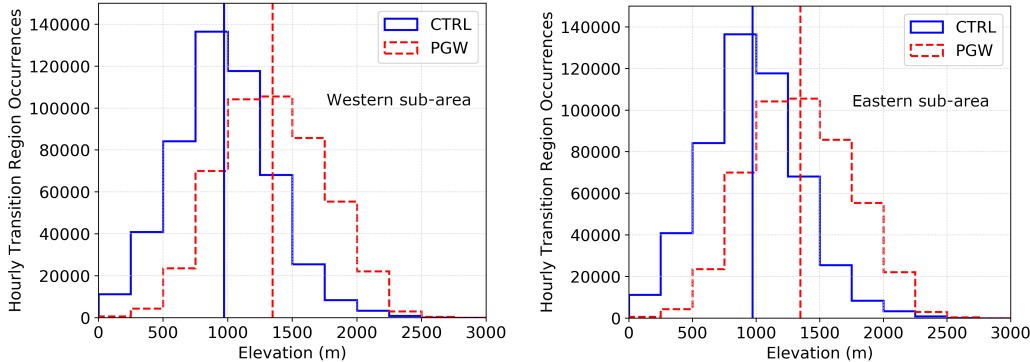

**Figure 10.** Transition region occurrences for the (a) western and (b) eastern sub-areas binned according to elevation. Vertical solid blue lines and dashed red lines represent the average elevation for CTRL and PGW, respectively.

The transition region categories also displayed variations in elevation (Fig. 11). In CTRL, the average elevations of categories with rain in the western sub-area were approximately 1,000 m, and categories with freezing rain were approximately 1,300–1,500 m in elevation except for freezing rain–snow–graupel which was lower, close to 1,000 m. In contrast, transition regions with rain in the eastern sub-area (at approximately 1,500 m) had the highest average elevation. Rain–graupel showed a dramatic difference in average elevation (approximately 830 m) between

the western and eastern sub-areas that was also much greater than one standard deviation. There were large ranges in actual elevations that were generally highest in the western sub-area, often over 2,000 m, whereas in the eastern sub-area the freezing rain-graupel category had a range of only 1,000 m. Although the ranges of elevation were large, the standard deviation of all categories were similar (approximately 250–450 m).

All categories over both sub-areas increased in average elevation, with the exception of rain–graupel over the western sub-area, which had a 15 m decrease (Fig. 11). Typical average elevation increases in the western sub-area were at least 300 m with a maximum of 447 m for the rain–snow–graupel category. Typical increases in the eastern sub-area were less (ranging from 32 m to 319 m), except for rain–graupel and freezing rain.





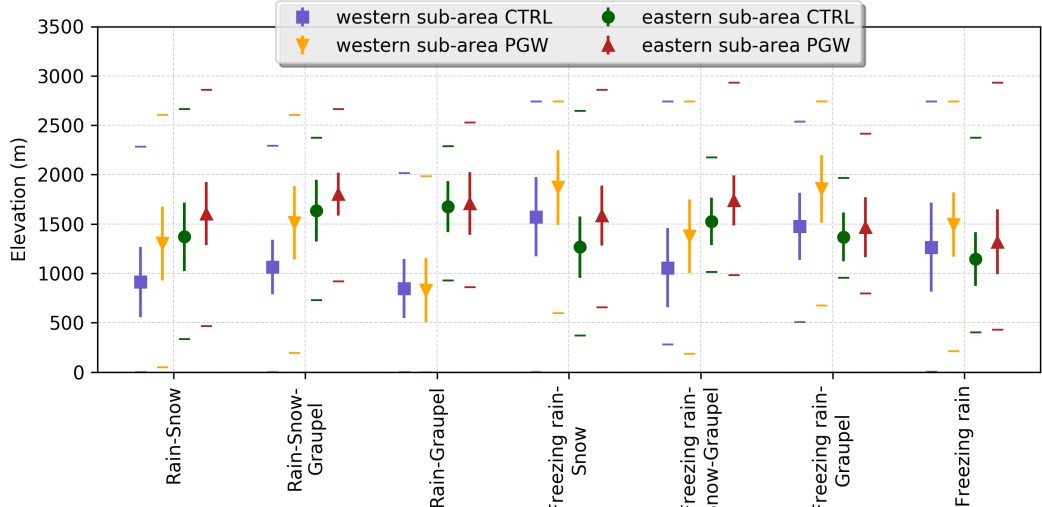

**Figure 11.** Average elevation of transition region categories over the western and eastern sub-areas under both the CTRL (green circles, purple squares) and PGW (red triangles, orange upside down triangles) and their standard deviations. Minimum and maximum elevations are denoted by horizontal bars.


As a consequence of vertical movements associated with higher temperatures, there were horizontal movements as well. As transition regions moved upwards, they also moved eastward from the lower Coast Mountains to the higher Rocky Mountains. This was quantified through a count of hourly transition regions categories across the western and eastern sub-areas (Table 3). Under PGW there was a 21,735 (4 %) decrease in the number of hourly transition region

occurrences across the western sub-area, in contrast to an increase of 177,282 (125 %) over the eastern sub-area. The fraction in the western sub-area consequently dropped from 77.8 % down to 59.8 %. This horizontal movement is similar to the findings of Klos et al. (2014) whereby transition regions in the Western United States are predicted to shift upwards in elevation and consequently eastward, where elevations are higher, but also northward in latitude, where temperatures are colder.


### 5.3 Extreme events

Extreme events were selected as the days within the top fifth percentile in terms of total transition region precipitation accumulation (Table 5). This led to six days being classified as extreme with three occurring during a

multi-day event on 11–15 January.

To characterize the associated synoptic patterns, six hourly Japanese 55 year reanalysis (JRA-55) data (Kobayashi et al., 2015) for mean sea level pressure (MSLP) and integrated water vapour transport (IVT) were analyzed.  IVT values were used as a proxy to determine whether atmospheric rivers were present.  Specifically, when IVT values

>250 kg m$^{-1}$ s$^{-1}$ made landfall, an atmospheric river was considered present.




All the extreme days were characterized by an Aleutian or Coastal low synoptic pattern and associated with a low level jet and concentrated corridors of water vapour (Fig. 12). These long and relatively narrow corridors have been documented as early as the 1970's within the United Kingdom (Browning and Pardoe, 1973), but more recently

referred to by Zhu and Newell (1992) as atmospheric rivers. In this study atmospheric rivers are characterized by long narrow filaments of water vapour, often preceding the cold front of a storm (Zhu and Newell, 1992). These atmospheric rivers have resulted in extreme precipitation and flooding over the Pacific Northwest, and although they have not been explicitly linked to transition regions, they have been attributed to rain-on-snow events (Guan et al., 2010; Radic et al., 2015). Their role in extreme transition region occurrences is linked to the warmer conditions

raising the melting elevation as well as enhancing moisture for precipitation production (Guan et al., 2010).

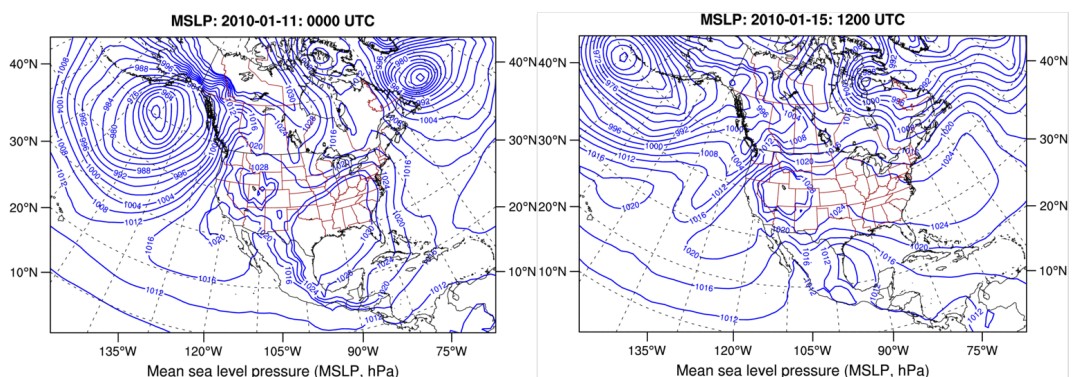

**Figure 12.** Mean sea level pressure using data from the JRA-55 to characterize synoptic patterns. (a) is an example

of an Aleutian Low occurring on January 11 0000 UTC and (b) is an example of a Coastal Low occurring on January 15 1200 UTC. These atmospheric rivers reached into the eastern sub-area for all of the extreme days, with the exception of 14 February. Under PGW, these five atmospheric rivers extended even farther east and with higher water vapour fluxes.

Previous research has found that atmospheric rivers can extend far inland. There are several trajectories whereby they enter the eastern sub-area but most occur when first making landfall over Oregon or Washington states (Rutz et al., 2015) and was found in each of the present events.

Some atmospheric rivers under PGW led to a decrease in transition region precipitation. Decreases of 11,434 mm

and 18,689 mm were found on 11 January and 27 April, respectively. Temperatures on these two days were so high over the Insular Mountains that more precipitation fell as liquid only.

**6 Implications for avalanches and ski resorts**


**6.1 Avalanche implications**





The formation of weak layers in the snowpack is essential for its instability and eventual avalanche production (COMET, 2010). Avalanches are complex slope failures and do not always occur naturally, but can be triggered by
human related activities. Therefore, this study only examines some of the atmospheric conditions that are conducive to the formation of weak layers.

Avalanches reported within four days of the top 5 % of transition region precipitation accumulation days were examined (Table 4). This four day criterion is similar to that used by Hatchett et al. (2017), whereby delayed
fatalities caused by avalanches were linked to atmospheric rivers. Given the importance of atmospheric rivers to the occurrence of large transition region events, this criterion was used. There were at least 21 avalanche incidents associated with the top 5 % transition region events with the majority (17) occurring in the eastern sub-area.

Several factors can contribute to avalanches that are also associated with transition regions. These are briefly
examined here with a focus on how they may change in the future. One factor is their mere presence. These regions can lead to weak layers, typically resulting in the release of direct-action avalanches, which was found in a study by Fitzharris (1976) in the Mt. Cook region of New Zealand. The stickiness of the wet snow found within many transition regions can allow it to adhere onto steep surfaces, which are not often avalanche prone, because snow sluffs off (Fitzharris, 1976).

A second factor is the increase in the number of transition region occurrences under PGW, which would imply more conducive environments for avalanches in the future. Ski resort statistics across the study area were retrieved through https://www.onthesnow.com/canada/statistics.html to determine where the average transition region elevation would lie with respect to the ski resorts; this includes all the ski resorts within the Alberta and BC ski
industry associations found in the study region. Under PGW, transition region occurrences would occur more frequently within 16 of the 26 ski resorts whereby avalanches could be human-triggered if no avalanche mitigation efforts were made (Fig. 13).




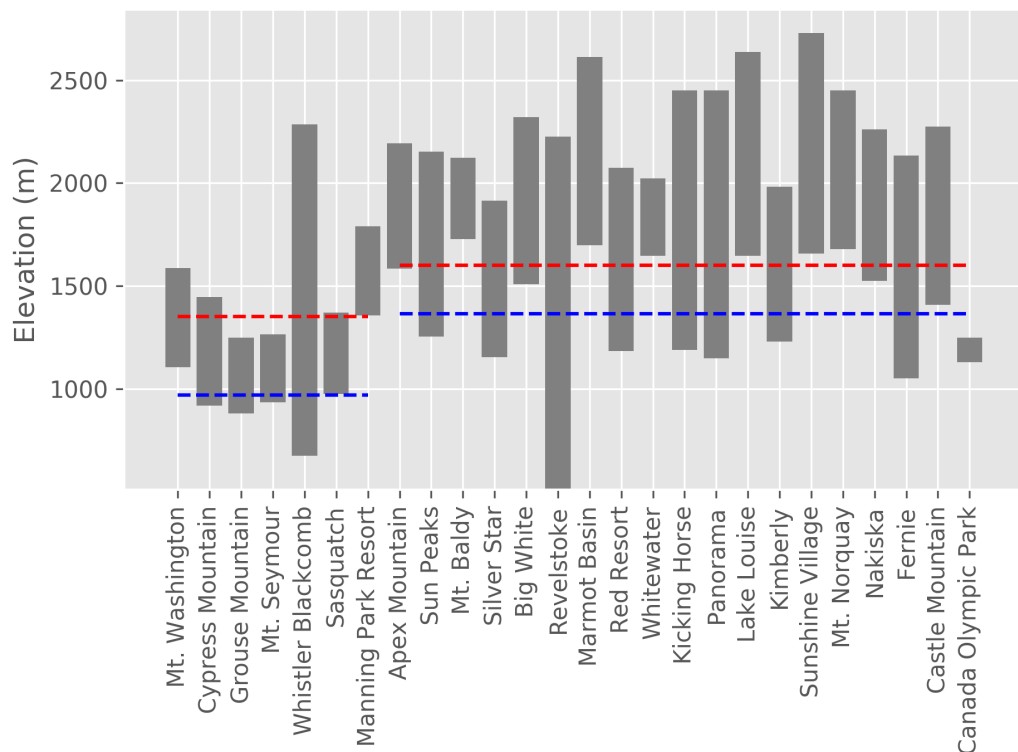

**Figure 13.** Elevation ranges (MSL) from ski resort bases to their mountain summits (retrieved from https://www.onthesnow.com/canada/statistics.html) and average elevations of transition regions under CTRL and PGW simulations. Ski resorts are ordered from west to east. The dashed blue and dashed red lines represent the four month average transition region elevations under the CTRL and PGW simulations, respectively, in the west and east sub-areas that are separated at 121° W.

A third factor, sometimes, is the presence of heavily rimed particles including graupel. These particles within transition regions are associated with weak layers (Lachapelle, 1967; McClung and Schaerer, 2006). Graupel, which does not adhere well to the surface due to its somewhat spherical nature, acts as a sliding layer within the snowpack for subsequent snowfall (Lachapelle, 1967; COMET, 2010). Accumulated graupel within the transition region under PGW increased by 28%, which implies increased avalanche risk in the future.

A fourth factor is the order of precipitation occurrence. For example, storms that start with predominantly snow and end with predominantly rain form a weak layer that is both unstable and acts as a sliding layer. An unstable layer is initially formed when heavier density precipitation overlies a layer of lighter density precipitation (McClung and Schaerer, 2006; COMET, 2010; Stimberis and Rubin, 2011).





The order of precipitation occurrence be related to atmospheric rivers. First of all, a simple conceptual model of the associated precipitation evolution can be envisioned. Prior to the landfalling atmospheric river, temperatures are cool and precipitation would begin as snow, followed by increasing temperatures with the passage of the warm

front, resulting in more liquid precipitation. As the cold front sweeps through, temperatures would drop and precipitation would fall as snow. The implication to this precipitation ordering would be the formation of rain crusts on the snowpack, and denser precipitation overtop less dense precipitation creating a layer of instability. The newly loaded snow associated with the cold front passage is then prone to avalanching.

This expected temporal evolution was considered beginning on the day preceding the 6 extreme days and ending the day following (Table 4). This precipitation evolution at ski resorts is shown for 14–16 January, as an illustrative example in Fig. 14. This idealized situation is evident at Whistler Mid Station, on 15 January, when snow began at 1700 UTC, warmed and turned into a transition for a couple of hours, and then cooled and accumulated as snow. Although the atmospheric river's trajectory extended into the eastern sub-area, the warmest temperatures did not

reach 0° C over its associated resorts so snow mainly occurred.

In general, the precipitation evolution was consistent with expectations over three of the six days (12 January, 15 January, 14 February) in the western sub-area. In the eastern sub-area, for reasons mentioned above and with ski resorts located at higher elevations, predominantly snow occurred.





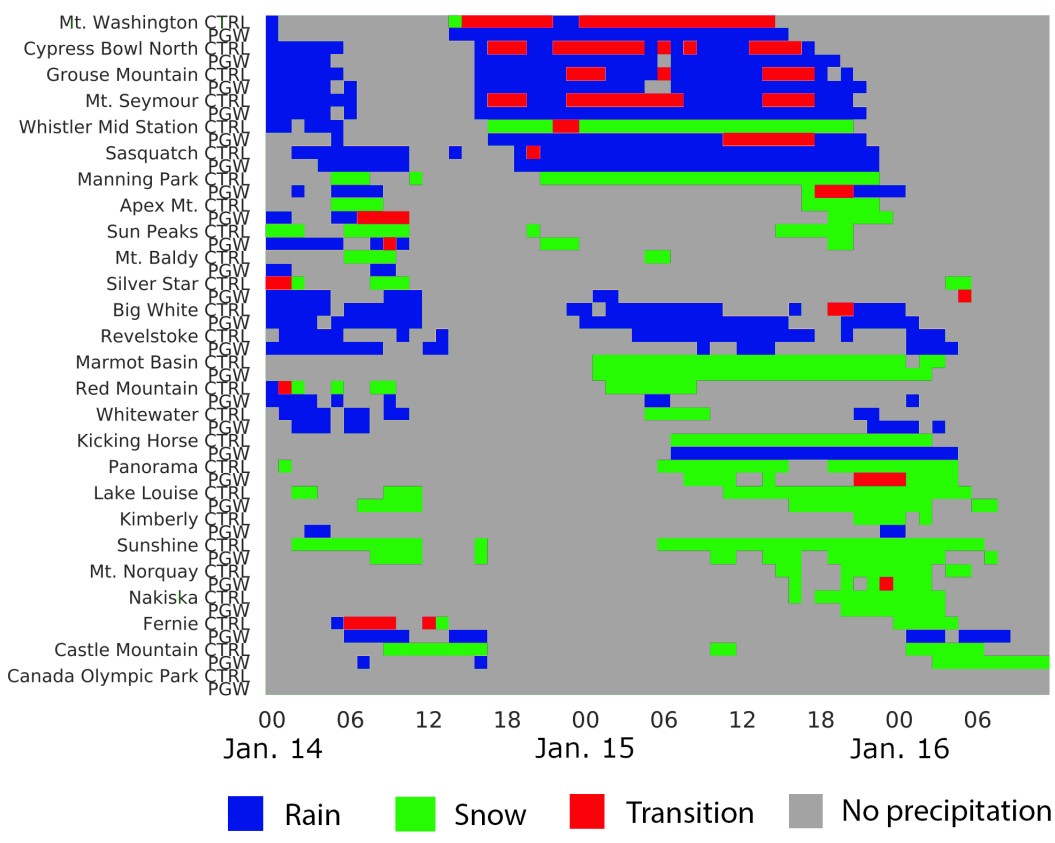

**Figure 14.** Timeline of precipitation type under CTRL and PGW at ski resort base from 14 January 0000 UTC, 2010 to 16 January 1200 UTC, 2010. Precipitation occurrence is shown as rain (blue), green (snow), red (transition region) and grey (no precipitation). Ski resorts are organized from west (top) to east (bottom).

Not all extreme days followed the exact precipitation evolution from snow to rain to snow. For example, on 29 March, at the lower elevation ski resorts the initial conditions were too warm, and only rain fell, hence skipping the first step of the evolution. The rest of the evolution occurred as expected. Furthermore, the orography and the diurnal cycle contributed to the complexity of the evolution. Often, rain would occur in the low lying valleys, whereas snow would occur at the peak. The precipitation evolution from snow to rain to snow would occur midway along the slope.

Another subtlety was the change between rain and snow without a transition region. A transition region must occur, but did not meet the conservative criteria over the hourly time scale (Sect. 2.2). An example occurs on 14 January at 1700 UTC, at Cypress Bowl North, Grouse Mountain and Mt. Seymour when precipitation switched immediately from all snow to all rain (Fig. 14).



The temperature evolution under PGW remained the same for the 14–16 January extreme event, but with higher temperatures. Therefore, snow that occurred under the CTRL across the western sub-area accumulated as either

transition region precipitation, or rain, whereas transition region precipitation under the CTRL accumulated as rain under PGW. However, at high elevation resorts such as Sunshine and Lake Louise only snow accumulated on both days since the melting level never rose that high.

## 6.2 Ski resort implications


Changes in the transition region would impact other sectors of the mountainous region's economy. Ski resorts are major economic drivers with 8.4 million visitors and $790 million in revenue over Western Canada in 2015/2016 (Nicolson, 2016). Ski resorts have been resilient to a changing climate, with the use of snow-making machines, but these are energy intensive, costly to run and require specific weather conditions to manufacture, therefore they are

not always a viable option.

With increasing elevations of the transition region, rain will be observed higher up the mountain basin (Fig. 13). This will raise the snow base and create more crust like layers which are undesirable for snow sport enthusiasts (McClung and Schaerer, 2006). Lower lying ski resorts, often found in the western sub-area, would be the most

impacted.

## 7 Concluding remarks

This study focussed on changes to the transition regions over the southern Canadian Cordillera from the Coast to the Rocky Mountains (115–127° W and 49–53° N) that would be associated with a warmer and more moist climate. The research exploited a number of observational and model datasets over the January–April 2010 period including the

pseudo-global warming (PGW) dataset (Liu et al., 2016). An evaluation of the control (CTRL) temperature and precipitation found that they compared reasonably well to the limited station observations and CaPA precipitation dataset and the region of focus is adjacent to well studied American regions using the same dataset (Liu et al., 2016). This article focussed on the transition region under both the CTRL and PGW simulations and has led to a number of insights as summarized below.


Transition regions were a prominent feature across the southern Canadian Cordillera. They occurred on 93 % (94 %) of the days under the CTRL (PGW). The number of hourly transition occurrences increased greatly with PGW (24 %); and their accumulated precipitation accounted for 12 % (13 %) of the total accumulated precipitation over the entire region under CTRL (PGW).


Transition regions were broken down into seven categories based on constituent precipitation types. All occurred within both the CTRL and PGW simulations. The rain–snow category was most prevalent under both the CTRL and



PGW simulations, and within both sub-areas of the region of focus (separated at  121° W) followed by the rain–snow–graupel category, whereas freezing rain was least common.


Overall warming contributed to an upward movement of the transition region that varied between the sub-areas and was additionally aided by elevation–dependent warming. Under PGW, the average transition region elevation increased from 971 ±380 m to 1,351 ±380 m for the western sub-area and from 1,365 ±346 m to 1,600 ±326 m. These elevations align with the greatest temperature increases that occurred in the 700–1,700 m layer with the loss

of snow coverage and subsequent surface albedo reduction (Mountain Research Initiative EDW Working Group, 2015).

There were differences in transition region occurrences between the western and eastern sub-areas.  Transition regions over the western sub-area decreased by 4 % as the melting layer more frequently occurred above the

topography, and they increased over the eastern sub-area by 125 %, a consequence of temperatures becoming warm enough to allow transition regions and the higher terrain ensuring that the melting layer was not above it.

The top 5 % of days producing the most transition region precipitation (6 days) displayed similar characteristics. They were all associated with land-falling atmospheric rivers, which brought higher temperatures and more

moisture. The expected precipitation evolution of snow, transition region, rain and snow did not always occur however, due to variations between atmospheric rivers as well as the terrain over which they occurred.

Findings from this study have important implications. The increase in transition region precipitation, more graupel occurrences and a higher proportion of rain within the transition region would all contribute to the destabilization of

the snowpack, increasing the risk of avalanches. The expected increase in the average elevation of the transition region, will change the areas at risk for avalanching towards higher terrain. Furthermore, this upward movement will impact ski resorts and their operations, particularly those over the western sub-area that are lower in terrain.

It is recognized that large scale dynamic change was not considered and would have major consequences on the

movement of storm tracks in scenarios of future conditions. A follow-on study should consider, for example, changing dynamics over a longer time period and a larger study area that includes the entire Canadian Rocky and Coast Mountains Ranges.

In summary, transition regions were almost always present over the southern Canadian Cordillera over the four

month illustrative period of January–April 2010. Under a PGW assumption, it is expected that transition regions will increase in hourly occurrence, duration, elevation and precipitation amount with more freezing rain and graupel while generally moving eastward. Implications would affect many aspects of society, hydrology and ecology.



**Data Availability**

The WRF HRCONUS and JRA-55 datasets are available through the following URLs, respectively:

https://rda.ucar.edu/datasets/ds612.0/, https://rda.ucar.edu/datasets/ds628.0/ (registration required).

**Author Contribution**

JA was the lead author for this manuscript and carried out the scientific analyses. RS wrote part of the manuscript, assisted with suggestive edits and provided scientific expertise into the analyses.

**Competing Interests**

The authors have no competing interests.

**Disclaimer**

**Special Issue Statement**

This article is part of the special issue "Understanding and predicting Earth system and hydrological change in cold regions". It is not associated with a conference.

**Acknowledgements**

The authors would like to thank the National Center for Atmospheric Research for providing the WRF dataset that made this article possible and especially Kyoko Ikeda for helping to facilitate data transfer. Utmost gratitude is given to Dr. Ruping Mo who provided products using the JRA-55 dataset that were used for synoptic analyses. The research was supported by the Changing Cold Regions Network supported by the Natural Sciences and Engineering

Research Council (NSERC), the Global Water Futures programme supported by Canada First Research Excellence Fund, and the NSERC Discovery grant of Ronald Stewart.





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





**Table 1**: Transition region categories and their associated precipitation types. Check marks indicate when the $\geq 0.2$ mm criterion was satisfied. A temperature criterion $\leq 0°$ C was used to exclude rain, whereas a criterion $> 0°$ C was used to exclude freezing rain.

| Transition Region Category | Precipitation Type | | | |
| --- | --- | --- | --- | --- |
| | Snow | Rain | Graupel | Freezing Rain |
| Rain–snow | ✓ | ✓ | x | x |
| Rain–snow–graupel | ✓ | ✓ | ✓ | x |
| Rain–graupel | x | ✓ | ✓ | x |
| Freezing rain–snow | ✓ | x | x | ✓ |
| Freezing rain–snow–graupel | ✓ | x | ✓ | ✓ |
| Freezing Rain–Graupel | x | x | ✓ | ✓ |
| Freezing Rain | x | x | x | ✓ |

**Table 2**. ECCC observation stations used for CTRL temperature and relative humidity verification for the January–
April 2010 period. Stations are listed from west to east. Meteorological data are averaged over the four month period at each ECCC station and at the closest CTRL model grid point. RH refers to relative humidity and the * symbol refers to information only available from 0700 to 1700 local time at Golden.

| Comparison of Temperature and Relative Humidity at ECCC Stations | | | | | | |
| --- | --- | --- | --- | --- | --- | --- |
| Station | ECCC Station Elevation (m) | WRF Elevation (m) | ECCC Station Temp. (°C) | WRF Temp. (°C) | ECCC Station RH (%) | WRF RH (%) |
| Cypress Bowl North | 953 | 729 | 1.8 | 3.6 | 85 | 75 |
| Whistler Mid Station | 1320 | 1133 | -0.3 | 0.0 | 86 | 69 |





| | | | | | | |
|---|---|---|---|---|---|---|
| Revelstoke | 445 | 592 | 2.6 | 5.1 | 82 | 55 |
| Jasper Warden | 1020 | 1 226 | -1.2 | -0.8 | 70 | 55 |
| Glacier NP Rogers Pass | 1330 | 1860 | - | - | - | - |
| Golden | 785 | 922 | 2.3* | 2.0 | 72* | 50 |
| Yoho NP | 1320 | 1736 | - | - | - | - |
| Banff | 1397 | 1580 | -1.5 | -2.9 | 66 | 61 |
| Fernie | 1001 | 1199 | - | - | - | - |






**Table 3.** Number of hourly occurrences under each transition region category, along with their differences over both the western and eastern sub-areas, separated at 121° W. Occurrences in each category are shown as a proportion in their respective sub-areas and under PGW over do not add up to 100 %, due to rounding. Average elevations and total sum of occurrences are shown for the western and eastern sub-areas.

| Transition Category | Western Sub-area | | | | | Eastern Sub-area | | | | |
|---|---|---|---|---|---|---|---|---|---|---|
| | Number of hourly occurrences | | Difference (PGW-CTRL(h/%)) | Proportion (%) | | Number of Hourly Occurrences | | Difference (PGW-CTRL(h/%)) | Proportion (%) | |
| | CTRL | PGW | | CTRL | PGW | CTRL | PGW | | CTRL | PGW |
| Rain–Snow | 348,206 | 326,404 | -3,986 / -1 | 70.1 | 68.7 | 120,374 | 268,245 | 147,871 / 123 | 84.6 | 84.0 |
| Rain–Snow–Graupel | 63,255 | 68,024 | 4,769 / 8 | 12.7 | 14.3 | 875 | 10,863 | 9,988 / 1,141 | 0.6 | 3.4 |
| Rain–Graupel | 18,209 | 25,956 | -7,747 / -48 | 3.7 | 5.5 | 242 | 1,781 | 1,539 / 636 | 0.2 | 0.6 |
| Freezing Rain–Snow– | 27,888 | 20,308 | -7,580 / -27 | 5.6 | 4.3 | 11,763 | 15,989 | 4,226 / 36 | 8.3 | 5.0 |
| Freezing Rain–Snow–Graupel | 7,295 | 6,023 | -1,272 / -17 | 1.5 | 1.3 | 240 | 756 | 516 / 215 | 0.2 | 0.2 |
| Freezing Rain–Graupel | 1,529 | 1,024 | -505 / -33 | 0.3 | 0.2 | 71 | 489 | 418 / 589 | < 0.1 | 0.2 |
| Freezing Rain | 30,565 | 27,473 | -3,092 / -23 | 6.1 | 5.8 | 8,640 | 21,364 | 12,724 / 147 | 6.1 | 6.7 |
| | 496,947 | 475,212 | -21,735 / -4 | | | 142,205 | 319,487 | 177,282 / 125 | | |





**Table 4.** Events within the top fifth percentile over the January–April, 2010 period in terms of total transition region precipitation accumulation in the CTRL and PGW simulations.

| Date | Synoptic Type | CTRL Precipitation Accumulation (mm) | PGW Precipitation Accumulation (mm) | PGW-CTRL Difference (mm) |
|---|---|---|---|---|
| Jan 11 | Aleutian Low | 73,117 | 61,683 | -11,434 |
| Jan 15 | Coastal Low | 59,964 | 99,626 | 39,662 |
| Jan 12 | Coastal Low | 52,013 | 52,699 | 686 |
| Mar 29 | Aleutian Low | 51,246 | 100,708 | 49,462 |
| Apr 27 | Aleutian Low | 46,455 | 27,766 | -18,689 |
| Feb 14 | Aleutian Low | 46,022 | 49,428 | 3,340 |


