# Peer review of "Precipitation Transition Regions over the Southern Canadian Cordillera during January–April 2010 and under a Pseudo-Global Warming Assumption"

_Hydrology and Earth System Sciences, 2019_

## Referee Comment (RC1) · Anonymous Referee #1 · 28 Mar 2019

**Overview**

This manuscript evaluates winter precipitation in southwestern Canada, with a particular focus on mixed precipitation in a transition zone, using two simulation outputs from the Weather Research and Forecasting (WRF) model - control and pseudo-global warming (PGW). This research evaluates transition, or mixed, precipitation during a time period that includes the 2010 SNOW-V10 project. The results are discussed relative to changes in transition elevation and placed in the context of the ski industry. This research examines an important hydroclimatological feature that has implications for

transportation and recreation sectors and snowpack integrity in general.

General comments

The manuscript is well-written, but lacking some key elements. In particular, very little information on the WRF model or PGW are given. It is unclear how the WRF model classifies precipitation phase or how the transition regions are defined. Add a few sentences describing the model itself. Add a few sentences about what a PGW simulation is and how it differs from traditional climate change scenarios, or a broad overview of what the output of PGW simulations. A sentence or two describing the original study by Liu et al. would be beneficial.

The phrase "transition regions" is used throughout the paper; however, I assume these regions are defined by the 4 km grid points, therefore, the phrases "transition precipitation" or "transition grids" would be more appropriate. Additionally, it should be clarified if a transition region refers to a larger spatial area where mixed precipitation occurs or if a transition region refers to a single grid in which mixed precipitation occurs.

Given the links between this research, the SNOW-V10 project, and the ski industry, there should be a comparison between the WRF CTRL model output and the results of the SNOW-V10 project. Very little reference to the SNOW-V10 project was given beyond the Introduction. The authors should revise the manuscript to include comparisons and discussion between this research and the SNOW-V10 project.

Specific comments

Line 21: links to impacts to avalanche activity and ski resorts seems like an afterthought, add more context, given the extent of the links to these impacts in the manuscript

Line 35: references to avalanche activity in southwestern Canada would be more appropriate for this sentence. There are many studies, particularly for the Columbia Mountains

Lines 36-37: would be useful to discuss transition regions as variable in time and space

Line 39: might be more appropriate to use "transition precipitation" rather than "transition regions"

Lines 93-96: Need more context for the importance of transition regions to society, including more on ski resorts, as this is one of the foci of the study, and as an indicator of climate change

Line 90: HRCONUS dataset is not discussed further in Section 2, there is only a quick mention of the boundary between this study and HRCONUS. What is the relevance of HRCONUS to the present study?

Lines 136-140: With respect to "substantial precipitation" is there a minimum daily accumulation or just 0.2 mm per hour? Because if there was only one recorded precipitation event at that grid point for a single day, and the total accumulation was 0.2 mm, it is not considered a substantial amount. Also, substantial is a subjective word. Perhaps revise to state that the total daily accumulation may be higher than the minimum ECCC standard. This model may underestimate the total number of occurrences.

Lines 143-144: it is unclear how the categories were defined, or how the WRF model delineates precipitation phases. What are the "additional steps" to categorize precipitation?

Table 1: unnecessary. It is intuitive by the transition region category that certain precipitation types are included within the category. For example, it is intuitive that rain-snow includes rain and snow but not graupel and freezing rain, where rain-snow-graupel includes rain, snow, and graupel but not freezing rain.

Line 164: A margin of error is given for the resulting temperature from the WRF simulation, but what is the margin of error for precipitation phase? Considering the "difficulty of forecasting for precipitation types within the transition region" it seems pertinent to discuss potential errors or precipitation misclassification from the model output. Was

there any validation of precipitation phase done as there is for temperature? Would the overestimation or underestimation of temperature at given locations influence the resulting precipitation classifications?

Lines 355-363: This paragraph is poorly explained. The sentence "...when warm moist Pacific air entered the study area, the elevation of the 0°C isotherm would at times occur above the peaks of the Insular and Coast Mountains, effectively lowering the average elevation of the transition region" is unclear with respect to how the isotherm affects the elevation of the transition. Additionally, by "number of transition regions" are you referring to the number of grid points? If yes, then be explicit, perhaps refer to them as grid points with mixed precipitation or similar.

Figure 14: the y-axis is squished and hard to distinguish between the CTRL and PGW lines

---

## Referee Comment (RC2) · Anonymous Referee #2 · 28 Mar 2019

**General Comments:**

The study compares precipitation transition regions over Southern Canadian Cordillera during in January-April 2010 using high-resolution Weather Research Forecasting model simulations of retrospective and future climate. They identified regions and times of mixed-phase precipitation from the control and future simulations and group them into seven types of transition region based on constituent of precipitation. Area extent, distribution and elevation of transition regions are determined and compared between the historical and future simulations. They also investigate occurrence of each transi-

Printer-friendly version

Discussion paper

tion region type and amount of precipitation from the two simulations to explore future change in transition region features. The study examines causes to the changes in future climate runs and suggests potential increase in avalanche risks at major ski resorts.

In general, the manuscript is very well written and focused. The objective of the study is clearly stated. Analysis techniques is well explained. Observational data, both point and gridded sources, are used to first validate model performance in simulating temperature, humidity, and precipitation in the study area which is essential for achieving the study goal to investigate future changes. It is interesting to find out how a warmer climate impact not only the elevation difference of where transition regions would typically occur but also spatial extent and distribution.

Some errors in the figures (see below) were found and some of them could use modifications.

I believe after minor revisions this manuscript may be accepted.

Specific Comments: Abstract, line 13: change “reanalysis” to “reanalysis-driven” or delete the word.

Sec. 2.2, first paragraph: I recommend specifically noting here that Thompson one-moment microphysics scheme was used in the WRF simulations and rain, snow, and graupel are produced from the scheme (i.e., no ice pellets). Here, the authors wrote accumulation at the lowest level was used. Could you clarify if mixing ration of graupel, rain, and snow (that are available as 3D data) were used in your study or surface graupel, rain, and snow. If the lowest model height mixing ratios were used, please indicate how far above the ground (in general) the level is.

Sec. 2.2, last paragraph: Ice pellets can exist in transition regions. Did you consider situations with ice pellets? If so how did you determine ice pellets from the model outputs? If it was explained in Sec. 2.2 that only rain, snow, and graupel categories

Printer-friendly version

Discussion paper

are simulated in the mp scheme, then other readers may not wonder about how ice pellets were dealt with in this study.

Table 1: change “temperature criterion...” to “Wet bulb temperature criterion...”.

Sec. 3.1, line 156: Change “precipitation availability” to “precipitation observation availability”.

Figure 3: Panel (a) is said to be showing the simulation result mapped over a coarser 10km x 10km grid, but it looks like it is on a 4 km x 4 km grid space. Could you verify if a correct figure was used? The blue color scale makes it difficult to see where values are high/low. I would suggest using a different color scheme.

Sec. 3.2.1, line 202: It is good that the authors indicate issues with the gridded data product. However, should “>5mm” be “

occurrence can(is?) related to..."

Sec. 6.1, line 508: Change "This idealized ... at Whistler Mid Station ..." to "... at Whistler Mid Station CTRL ..."

Sec. 6.2, Figure 14: It is unclear from this section and Fig. 14 that the future climate will increase avalanche risk. Figure 14 shows that less transition regions at ski resorts in PGW which contradicts Lines 471-477 and analysis given prior to this section where clearly states an increase in transition region in PGW. Please clarify. Also, Figure 14 may be modified with horizontal lines separating each ski resort so that CTRL vs PGW comparison at a site can be done easily. Also indicate which ski resorts are on the eastern / western sub-regions or use separate panel plot for each sub-region.
* * *
Printer-friendly version

Discussion paper

---

## Author Comment (AC1) · 3 Aug 2019

We thank reviewer 1 for their critical and helpful comments. Below we address each one.

Reviewer 1

Comment: In particular, very little information on the WRF model or PGW are given. It is unclear how the WRF model classifies precipitation phase or how the transition regions are defined. Add a few sentences describing the model itself. Add a few

sentences about what a PGW simulation is and how it differs from traditional climate change scenarios, or a broad overview of what the output of PGW simulations. A sentence or two describing the original study by Liu et al. would be beneficial.

Response: These are good suggestions. We have revised the text as suggested.

Revised Text: "The WRF model is a numerical weather prediction model used for forecasting and research applications (Powers et al., 2017). In recent years, advancements in computing power have allowed for higher resolution simulations over regional scale areas.

This WRF simulation conducted by Liu et al. (2016) used 4 km resolution and explicit calculation of convection over the 2000-2013 period. This model was configured using the Thompson aerosol-aware microphysics scheme (Thompson and Eidhammer, 2014) with one-moment prediction of mass mixing ratio for cloud water, snow and graupel and two-moment prediction of the number concentration for cloud ice and rain. This scheme does not include ice pellets. These variables were then used as a basis for calculating the hourly surface snow, graupel and rain as discussed by Thompson and Eidhammer (2014). The determination of surface freezing rain requires a temperature criterion which is discussed in Section 2.2.

The actual simulations were carried out in two steps. First, a control (CTRL) run of the historical conditions using the European Centre for Medium-Range Weather Forecasts Re-Analysis (ERA)-interim was carried out followed by, second, a run using a Pseudo-Global Warming (PGW) assumption first used by Schär et al. (1996). This PGW method allows one to study the thermodynamic effects of a warmer, more moist climate on historical synoptic systems. In particular, this was accomplished by perturbing the CTRL with the Coupled Model Intercomparison Project 5 (CMIP5) (Taylor et al., 2012) ensemble mean high emissions scenario information at the end of the 21st century. This simulation addresses possible changes in storm intensity to historical storms within the latter twenty-first century (Liu et al. 2016). The PGW scenario varies from

traditional climate change scenarios as it is driven by reanalysis data but perturbed with a climate signal from 19 CMIP5 models. This approach does not consider changes in large scale forcing so it can largely be considered as examining the consequences of changes in thermodynamic forcing. This dataset was utilized in this article because of its high-resolution over complex terrain, availability and ease of access."

New references: Powers, J. G., Klemp, J. B., Skamarock, W. C., Davis, C. A., Dudhia, J., Gill, D. O., Coen, J.L., Gochis, D.J., Ahmadov, R., Peckham, S.E. and Grell, G. A.: The weather research and forecasting model: Overview, system efforts, and future directions. Bulletin of the American Meteorological Society, 98(8), 1717-1737, 2017.

Comment: The phrase "transition regions" is used throughout the paper; however, I assume these regions are defined by the 4 km grid points, therefore, the phrases "transition precipitation" or "transition grids" would be more appropriate. Additionally, it should be clarified if a transition region refers to a larger spatial area where mixed precipitation occurs or if a transition region refers to a single grid in which mixed precipitation occurs.

Response: Correct, transition regions were defined by grid points with accumulated mixed precipitation $\geq$ 0.2 mm within a 1 hour period. When this term is used, it refers to all the grid points that fulfill the definition. Therefore it can refer to a larger spatial area, or it can refer to just one single grid point (4x4 km region). We have changed the text so that spatial areas are referred to as transition regions and mixed precipitation is referred to as transition precipitation. We do not feel it is necessary to change the term "transition region" given a single grid point encompasses a larger spatial area and not a single point.

Revised Text: This change has been reflected in the manuscript as follows.

Lines 31-32 "For example, transition precipitation, such as freezing rain, can bring transportation to a halt on major highways such as happened on the Coquihalla Highway in British Columbia in 2017 (Canadian Press, 2017)."

Lines 39-41 "Transition precipitation across Canada has been studied for some time (Stewart and King, 1987; Stewart and Mcfarquhar, 1987; Stewart, 1992; Stewart et al., 1995; Cortinas et al., 2004; Theriault et al., 2012; Theriault et al., 2014; Groisman et al., 2016)"

Lines 63-64 "This campaign sought to improve winter weather forecasting within complex terrain, showcasing the difficulty of forecasting for transition precipitation (TheÌĄriault et al., 2014; Isaac et al., 2014)"

Lines 67-70 "During the SNOW-V10 period, the Olympic venues experienced several issues with warm weather, which led to delayed events over Cypress Mountain, whereas Whistler, at a higher elevation, received a great deal of snow, but also experienced several transition precipitation occurrences (Goldenberg, 2010; Guttsman, 2010; TheÌĄriault et al., 2012; Thériault et al., 2014; Isaac et al., 2014; Joe et al., 2014)."

Lines 133-155 "Specifically, within the model, this refers to grid points where there is transition precipitation at the surface on an hourly basis."

Lines 262-264 "However, the occurrence, spatial distribution and amount of transition precipitation varied greatly from the CTRL to the PGW and are discussed below."

Lines 281-282 "The total accumulated transition precipitation increased from 1,263,044 mm (CTRL) to 1,606,163 mm (PGW) a 27% increase and it accounted for ∼12% (13%) of the total precipitation under the CTRL (PGW)."

Lines 285-287 "This transition precipitation mainly occurred over the Insular and Coast Mountains under both the CTRL and PGW. However, under the PGW, less transition precipitation was simulated over the Insular Mountains and more over the Coast Mountains, especially its leeward side (Fig. 7)."

Lines 413-414 "Extreme events were selected as the days within the top fifth percentile in terms of total transition precipitation accumulation (Table 5)."

Line 444 "Some atmospheric rivers under PGW led to a decrease in transition precipitation."

Lines 534-536 "Therefore, snow that occurred under the CTRL across the western sub-area accumulated as either transition precipitation, or rain, whereas transition precipitation under the CTRL accumulated as rain under PGW."

Lines 585-586 "The expected precipitation evolution of snow, transition precipitation, rain and snow did not always occur however, due to variations between atmospheric rivers as well as the terrain over which they occurred."

Lines 588-590 "The increase in transition precipitation, more graupel occurrences and a higher proportion of rain within the transition region would all contribute to the destabilization of the snowpack, increasing the risk of avalanches."

Comment: Given the links between this research, the SNOW-V10 project, and the ski industry, there should be a comparison between the WRF CTRL model output and the results of the SNOW-V10 project. Very little reference to the SNOW-V10 project was given beyond the Introduction. The authors should revise the manuscript to include comparisons and discussion between this research and the SNOW-V10 project.

Response: Thanks for that comment. The purpose of our study was to look at an area far larger than that of SNOW-V10 and therefore did not focus on the results of it. We have recommended that the HRCONUS dataset be evaluated against the data collected during SNOW-V10 for follow on work. Also, one additional SNOW-V10 reference was added in the introduction that is relevant to this study.

Revised Text: This is the same revised text used in a response to another comment below. "A follow on study should consider a thorough evaluation of the HRCONUS dataset to precipitation type information. There are few manual stations that provide precipitation phase within the whole study area and 24 hours/day. However, some specialized automatic stations have recorded precipitation type over short time periods

with the use of an optical disdrometer such as during the 2010 SNOW V-10 experiment (Joe at al., 2014)."

New reference:

Joe, P., B. Scott, C. Doyle, G. Isaac, I. Gultepe, D. Forsyth, S. Cober, E. Campos, I. Heckman, N. Donaldson, D. Hudak, R. Rasmussen, R.E. Stewart, J.M. Thériault, H. Carmichael, M. Bailey and F. Boudala: The monitoring network of the Vancouver 2010 Olympics. Pure Applied Geoph., 171, 25-58, 2014.

Comment: Line 21: links to impacts to avalanche activity and ski resorts seems like an afterthought, add more context, given the extent of the links to these impacts in the manuscript

Response: We do not agree with these comments. There's a whole section dedicated to avalanche activity and the transition region. This was a critical motivation for the study and a driving force for the research and we have shown, for example, that atmospheric rivers have been linked to avalanche activity. We have made some small changes to the text in regards to ski resorts to better reflect these considerations but we feel that our existing text regarding avalanches is sufficient.

Revised Text: There is no new text regarding avalanches.

Revised text with regards to ski resorts "Precipitation occurring within the transition region often implies melting precipitation which is not ideal for skiers who prefer light density snow crystals. With increasing elevations of the transition region, rain will be observed higher up the mountain slopes (Fig. 13)."

Comment: Line 35: references to avalanche activity in southwestern Canada would be more appropriate for this sentence. There are many studies, particularly for the Columbia Mountains.

Response: Thank you for your comment. We have added additional material and references to Line 35, including studies by Jamieson and Langevin (2004) and Jamieson

et al. (2009) concerning avalanches in the Columbia Mountains. In particular they mention dry on wet faceting, whereby dry snow overlies wet snow is a factor in snow instability that can lead to avalanche occurrence.

Revised Text: "Moreover, avalanches occurring over southwestern Canada can disrupt traffic and cause injury or death to those in its path. Some of these avalanches can be attributed to precipitation within transition regions including its temporal ordering which can occur when wet snow associated with a warm front is followed by dry snow from a cold front (McClung and Schaerer, 1993; Abe, 2002; Abe, 2004; Jamieson and Langevin, 2004; Jamieson et al., 2009; COMET, 2010)."

New references: Jamieson, B., and Langevin, P.: Faceting above crusts and associated slab avalanching in the Columbia Mountains. Proceedings of the 2004 International Snow Science Workshop in Jackson Hole, Wyoming, USA. American Avalanche Association, 2004.

Jamieson, B., Haegeli, P. and Schweizer, J.: Field observations for estimating the local avalanche danger in the Columbia Mountains of Canada, Cold Regions Science and Technology, 58(1-2), 84–91, doi:10.1016/j.coldregions.2009.03.005, 2009.

Comment: Line 36-37: would be useful to discuss transition regions as variable in time and space

Response: Thank you, this is a good idea. Please see revised text.

Revised Text: "Transition regions are variable in time and in space. Their occurrence is dependent on many external factors including changing large scale atmospheric temperatures and moisture as well as variations in terrain. Their occurrence and features also vary due to internal factors related to, for example, cooling by melting and sublimation that can lead to transition regions moving horizontally (Stewart and McFarquhar, 1987) or down mountain slopes (Theriault et al., 2012; Stoelinga et al., 2012)."

Comment: Line 39: might be more appropriate to use "transition precipitation" rather

than "transition regions"

Response: Thank you, this has been done.

Revised Text: "Transition precipitation across Canada has been studied for some time (Stewart and King, 1987; Stewart and McFarquhar, 1987; Stewart, 1992; Stewart et al., 1995; Cortinas et al., 2004; Theriault et al., 2012; Theriault et al., 2014;Groisman et al., 2016)."

Comment: Lines 93-96: Need more context for the importance of transition regions to society, including more on ski resorts, as this is one of the foci of the study, and as an indicator of climate change.

Response: Lines 31-37 address the societal importance and the impact of climate change. We have added to this.

Revised Text: "Moreover, avalanches occurring over southwestern Canada can disrupt traffic, or cause injury or death to those in their paths. At least some of these avalanches can be attributed to transition precipitation and its sequential ordering (McClung and Schaerer, 1993; Abe, 2002; Abe, 2004; COMET, 2010). The ski industry is an important economic sector in many mountain towns, drawing both national and international tourists. These well-established ski resorts depend on the consistent seasonal occurrence of solid precipitation. Transition regions that occur within ski resorts demarcate poor skiing conditions and, to counteract this, additional maintenance is required to groom and maintain adequate skiing conditions which may include the use of snow making equipment."

Comment: Line 90: HRCONUS dataset is not discussed further in Section 2, there is only a quick mention of the boundary between this study and HRCONUS. What is the relevance of HRCONUS to the present study?

Response: We took advantage of a unique high-resolution dataset that was readily available and included information within Canada. Liu et al. (2016) found good correlation between the orographic precipitation over the western United States that was evaluated against United States precipitation datasets. As the HRCONUS includes our study region and geography is similar across the border, we took advantage of the large dataset. Moreover, this gave us an opportunity to evaluate the data against Canadian observations. We have included more text about the relevance of HRCONUS to our study.

Revised Text: "In particular, the National Center for Atmospheric Research (NCAR) has carried out 4 km resolution simulations using the Weather Research and Forecasting (WRF) model focussed over the contiguous United States but also including southern Canada and northern Mexico (Liu et al., 2016). The correlation was good, when model products were evaluated against United States precipitation datasets in the orographic regions of western United States. Given this success, and because our study area is just north of the United States border with similar terrain, we took the opportunity to analyze this unique dataset over Canada. These simulations focussed on a multi-year control period in the recent past (2000-2013) as well as over the same period but under warmer and more moist future conditions using a Pseudo-Global Warming approach. This high-resolution dataset over the contiguous United States (HRCONUS) is discussed in more detail in Sect. 2."

Comment: Lines 136-140: With respect to "substantial precipitation" is there a minimum daily accumulation or just 0.2 mm per hour? Because if there was only one recorded precipitation event at that grid point for a single day, and the total accumulation was 0.2 mm, it is not considered a substantial amount. Also, substantial is a subjective word. Perhaps revise to state that the total daily accumulation may be higher than the minimum ECCC standard. This model may underestimate the total number of occurrences.

Response: This is a good point. The only criterion used was the accumulation of at least 0.2 mm per hour. We have used your suggestion.

Revised Text: "Through this higher threshold, this study is examining transition regions with greater precipitation accumulation than the minimum ECCC standard and may consequently underestimate the total number of occurrences."

Comment: Lines 143-144: it is unclear how the categories were defined, or how the WRF model delineates precipitation phases. What are the "additional steps" to categorize precipitation?

Response: Thank you for your comment. We thought that we had explained this well but obviously not.

Revised Text: "The surface based categories were separated automatically using data products generated from the Thompson aerosol-aware microphysics scheme (Thompson and Eidhammer, 2014). The scheme predicts the mass mixing ratio of five explicit hydrometeor species: cloud water, cloud ice, rain, snow and a combined species of graupel-hail. These are predicted using equations that account for the microphysical processes that lead to each individual species including ice nucleation, deposition, sublimation, melting, accretion, autoconversion of species, etc. Number concentration is predicted for only cloud ice and rain."

Comment: Table 1: unnecessary. It is intuitive by the transition region category that certain precipitation types are included within the category. For example, it is intuitive that rain-snow includes rain and snow but not graupel and freezing rain, where rain-snow-graupel includes rain, snow, and graupel but not freezing rain.

Response: We appreciate your comments but we feel that Table 1 is useful. One can visually see the different precipitation types and combinations and better appreciate the complexity of these regions. This clarity is very useful to the reader later on when these are discussed individually.

Revised Text: There is no change in the text.

Comment: Line 164: A margin of error is given for the resulting temperature from the

WRF simulation, but what is the margin of error for precipitation phase? Considering the "difficulty of forecasting for precipitation types within the transition region" it seems pertinent to discuss potential errors or precipitation misclassification from the model output. Was there any validation of precipitation phase done as there is for temperature? Would the overestimation or underestimation of temperature at given locations influence the resulting precipitation classifications?

Response: This is a good point. Given the sparseness of the observations, it was not possible to carry out a rigorous evaluation of precipitation phase. This has been better articulated in the text including probable impacts of temperature simulations on precipitation phase.

Revised Text: "There will also be uncertainty in precipitation types as a result of the temperature uncertainties but this has not been quantified. For example, at four of the six stations where observed and model temperatures were compared, a warm bias was found. It may be that, at these locations there would be an accompanying tendency towards liquid forms of precipitation."

In concluding remarks and the same revised text as mentioned in a previous comment: "A follow on study should consider a thorough evaluation of the HRCONUS dataset to precipitation type information. There are few manual stations that provide precipitation phase within the whole study area and 24 hours/day. However, some specialized automatic stations have recorded precipitation type over short time periods with the use of an optical disdrometer such as during the 2010 SNOW V-10 experiment (Joe at al., 2014)."

New reference: Joe, P., B. Scott, C. Doyle, G. Isaac, I. Gultepe, D. Forsyth, S. Cober, E. Campos, I. Heckman, N. Donaldson, D. Hudak, R. Rasmussen, R.E. Stewart, J.M. Thériault, H. Carmichael, M. Bailey and F. Boudala: The monitoring network of the Vancouver 2010 Olympics. Pure Applied Geoph., 171, 25-58, 2014.

Comment: Lines 355-363: This paragraph is poorly explained. The sentence ". .

.when warm moist Pacific air entered the study area, the elevation of the 0°C isotherm would at times occur above the peaks of the Insular and Coast Mountains, effectively lowering the average elevation of the transition region" is unclear with respect to how the isotherm affects the elevation of the transition. Additionally, by "number of transition regions" are you referring to the number of grid points? If yes, then be explicit, perhaps refer to them as grid points with mixed precipitation or similar.

Response: When the 0°C isotherm rises significantly above the terrain, there is no longer a transition region at the surface since any precipitation would just be rain. The ensuing overall "number of grid points with transition precipitation" would be reduced and the overall average elevation (without these high elevation ones) would be lower. Your suggestion has been used and the text is more explicit.

Revised Text: " . . .when warm moist Pacific air entered the study area, the elevation of the 0°C isotherm would at times occur significantly above the peaks of the Insular and Coast Mountains so that only rain fell. The ensuing number of grid points with transition precipitation would be reduced and the overall average elevation (without these high elevation ones) would be lower..."

Comment: Figure 14: the y-axis is squished and hard to distinguish between the CTRL and PGW lines

Response: Thanks, the figure has been updated.

Revised Text: Please see attached figure

―――――――――――――――

[Figure]

**Fig. 1.** Figure 14. Timeline of precipitation type under CTRL and PGW at ski resort base from 14 January 0000 UTC, 2010 to 16 January 1200 UTC, 2010. Precipitation occurrence is shown as rain (blue), green (sn

---

## Author Comment (AC2) · 3 Aug 2019

We thank reviewer 2 for their critical and helpful comments. Below we address each one.

Comment: Abstract, line 13: change "reanalysis" to "reanalysis-driven" or delete the word.

Response: Good point.

[Figure]

Revised Text: The term "reanalysis-driven" is now used.

Comment: Sec. 2.2, first paragraph: I recommend specifically noting here that Thompson one moment microphysics scheme was used in the WRF simulations and rain, snow, and graupel are produced from the scheme (i.e., no ice pellets). Here, the authors wrote accumulation at the lowest level was used. Could you clarify if mixing ratios of graupel, rain, and snow (that are available as 3D data) were used in your study or surface graupel, rain, and snow. If the lowest model height mixing ratios were used, please indicate how far above the ground (in general) the level is.

Response: Good point. The surface graupel, rain and snow 2D outputs were used for this study and not 3D data. These surface 2D outputs take into account several microphysical processes, such as melting, evaporating, sublimation, deposition, and autoconversion.

Revised Text: Starting from line 108, the manuscript now reads: "Two 13 year high-resolution convective-permitting simulations from 2000 to 2013 were carried out by Liu et al. (2016) using the Weather and Research Forecasting (WRF) model version 3.4.1. This model was configured using the Thompson aerosol-aware microphysics scheme (Thompson and Eidhammer, 2014) with one-moment prediction of mass mixing ratio for cloud water, snow and graupel and two-moment prediction of the number concentration for cloud ice and rain. This scheme does not include ice pellets. "

Comment: Sec. 2.2, last paragraph: Ice pellets can exist in transition regions. Did you consider situations with ice pellets? If so how did you determine ice pellets from the model outputs? If it was explained in Sec. 2.2 that only rain, snow, and graupel categories are simulated in the mp scheme, then other readers may not wonder about how ice pellets were dealt with in this study.

Response: Thank you for your suggestion. Ice pellets were not considered in this study and this is now clarified in the manuscript. Concluding remarks have also been revised.

[Figure]

Revised Text: This has been adjusted and is the same as the revised text mentioned in an earlier comment: "This model was configured using the Thompson aerosol-aware microphysics scheme (Thompson and Eidhammer, 2014) with one-moment prediction of mass mixing ratio for cloud water, snow and graupel and two-moment prediction of the number concentration for cloud ice and rain. This scheme does not include ice pellets. These variables are then used as a basis for calculating the hourly surface snow, graupel and rain as discussed by Thompson and Eidhammer (2014). The determination of surface freezing rain requires a temperature criterion which is discussed in Section 2.2."

In concluding remarks: "Ice pellets were not considered in this study, but can certainly occur within transition regions, when an inversion of warm air aloft occurs. A future study may consider using the bulk microphysics scheme developed by Cholette et al. (2019) that explicitly predicts not only ice pellets but also wet snow.

New reference: Cholette, M., H. Morrison, J.A. Milbrandt, and J.M. Thériault: Parameterization of the bulk liquid fraction on mixed-phase particles in the Predicted Particle Properties (P3) scheme: Description and idealized simulations. J. Atmos. Sci., 76, 561–582, https://doi.org/10.1175/JAS-D-18-0278.1, 2019.

Comment: Table 1: change "temperature criterion. . ." to "Wet bulb temperature criterion. . .".

Response: Thank you, this has been done.

Revised Text: "A wet bulb temperature criterion $\leq 0^\circ$ C was used to exclude rain, whereas a criterion > $0^\circ$ C was used to exclude freezing rain."

Comment: Sec. 3.1, line 156: Change "precipitation availability" to "precipitation observation availability".

Response: Good point.

Revised Text: The phrase "precipitation observation availability" is now used.

Comment: Figure 3: Panel (a) is said to be showing the simulation result mapped over a coarser 10km x 10km grid, but it looks like it is on a 4 km x 4 km grid space. Could you verify if a correct figure was used? The blue color scale makes it difficult to see where values are high/low. I would suggest using a different color scheme.

Response: Thank you for your comment. We have verified that Panel (a) is on a 10 x 10 km grid and was regridded using a conservative interpolation using National Center for Atmospheric Research Command Line. We have also considered a different colour scheme but we feel the current one showing the gradient of precipitation accumulation is fine.

Revised Text: There is no revised text.

Comment: Sec. 3.2.1, line 202: It is good that the authors indicate issues with the gridded data product. However, should ">5mm" be "<5mm" according to Lespinas et al. (2015)? It may also be good to state that the density of observations used in generating CaPA drops significantly across US-Canada border as you go northward.

Response: Thank you for this comment. Precipitation events > 5mm are underestimated according to Lespinas et al. (2015). It is true that the density of observations decreases substantially, moving northward. Your suggestion has been added to the text.

Revised Text: One change was made as follows: "The density of stations used in generating CaPA drops significantly across the US-Canada border as one moves northward."

Comment: Sec. 3.2.2, line 220: Could you briefly explain how the data were adjusted by Mekis and Vincent (2011)?

Response: Environment and Climate Change Canada have deployed different types of precipitation measurement equipment and have conducted field experiments with the various precipitation gauges. They found greater accuracy with newer precipitation

gauges, therefore they employed different quality control techniques, dependent on the precipitation gauge variety including loss from evaporation, wetting loss and retention correction.

Revised Text: "Adjusted data by Mekis and Vincent (2011) were used where available and these include Glacier NP Rogers Pass, Golden and Fernie. Mekis and Vincent (2011) adjusted for errors in both rainfall and snowfall measurements. For rainfall measurements, rain-gauge specific corrections for three of the major gauge types were used within ECCC and each was adjusted for undercatch from wind, wetting at both the funnel and the receiver or container, and evaporation. Snowfall when measured by a ruler was converted to water equivalent by applying a snow water equivalent adjustment factor. This factor was determined by comparing gauge and snow ruler measurements following the techniques described by Metcalfe et al. (1994)."

New reference: Metcalfe, J. R., Ishida, S., and Goodison, B. E.: A corrected precipitation archive for the Northwest Territories of Canada. In Mackenzie Basin Impact Study, Interim Report# 2–Proceedings of the sixth biennial AES-DIAND meeting of Northern Climate and Mid Study Workshop of the Mackenzie Basin Impact Study, Yellowknife, Northwest Territories (Canada), 1994.

Comment: Figure 4: correct "WRF CCTRL" to "WRF CTRL" in the legend.

Response: Thank you for catching this. It has been fixed in the legend.

Revised Text: Please see attached figure.

Comment: Sec. 4, line 260: Sentence starting with "Of these, 93% (94%). . ." does not indicate which value is associated with the CTRL simulation. Correct the sentence accordingly.

Response: Thanks for pointing that out.

Revised Text: "Of these, 93% (94%) had a transition region occurrence under the CTRL (PGW) simulations."

Comment: Sec. 4-5: I may have missed it but there is no mention of how much relative humidity changed from CTRL to PGW simulations over the study area. This study does not mention about how change in cloud mixing ratio and vapor mixing ratio (in PGW) would impact evolution various hydrometeors. A short discussion on change in moisture (not only temperature) would be good.

Response: Good suggestion. We have included a discussion of change in relative humidity. Cloud mixing ratio and water vapor mixing ratio were not specifically examined in this study. However, it would be expected that under PGW there would be more water vapour and this could lead to more accretion and graupel.

Revised Text: "The CTRL relative humidities were lower (5–27%) when compared to the ECCC stations. The largest discrepancy was at Revelstoke particularly at the beginning of the study period from January to mid-March. For transition regions, these subsaturated surface conditions could mean that the model may be underestimating precipitation at the surface due to sublimational or evaporative losses and it could have an effect on the type of transition precipitation since the melting process is slowed."

"Although not considered here, it is expected that the higher water vapour content and would lead to more accretion and graupel."

Comment: Figure 10: The two panels are identical (western sub-region?). Please check.

Response: Sorry for our mistake.

Revised Text: The correct panels are now used and are attached.

Comment: Sec. 6.1, line 496: Correct "the order of precipitation occurrence be related to .." to " occurrence can(is?) related to. . ."

Response: Thank you for noticing that.

Revised Text: "The order of precipitation occurrence can be related to atmospheric

rivers."

Comment: Sec. 6.1, line 508: Change "This idealized . . . at Whistler Mid Station . . ." to ". . . at Whistler Mid Station CTRL . . . "

Response: Good suggestion.

Revised Text: "This idealized situation is evident at Whistler Mid Station CTRL..."

Comment: Sec. 6.2, Figure 14: It is unclear from this section and Fig. 14 that the future climate will increase avalanche risk. Figure 14 shows that less transition regions at ski resorts in PGW which contradicts Lines 471-477 and analysis given prior to this section where clearly states an increase in transition region in PGW. Please clarify.

Response: Thank you for pointing this out. Lines 471-477 refer to Figure 13 and refers to the entire duration of the study, whereas Figure 14 illustrates the order of precipitation type at each ski resort only over a few days (14-16 January), and is not representative of the 4 month period. Text for Figures 13 and 14 has been revised to clarify this.

New Text: Text in relation to Figure 13 now reads "Under CTRL the average 4 month transition region occurred within the elevation range of 12 ski resorts but, under PGW, this average occurred within the elevation range of 16 ski resorts (Fig. 13). Six additional ski resorts were added under PGW but two were lost because the average 4 month transition region height moved above them (Grouse Mountain and Mt. Seymour). These six additional ski resorts could experience more human-triggered avalanches if no avalanche mitigation efforts are made."

Text in relation to Figure 14 now reads "This precipitation evolution at ski resorts is shown for 14–16 January, as an illustrative example in Fig. 14. This figure is not an accurate representation of the entire study period (January–April 2010)."

Comment: Also, Figure 14 may be modified with horizontal lines separating each ski resort so that CTRL vs PGW comparison at a site can be done easily. Also indicate

which ski resorts are on the eastern / western sub-regions or use separate panel plot for each sub-region.

Response: We considered following this suggestion of examining each ski resort but did not feel that it is necessary. We are just trying to show the general pattern. We have also improved the figure somewhat to more clearly indicate ones on the eastern and western sides.

Revised Text: The new figure is attached and revised caption is below.

Figure 14: Timeline of precipitation type under CTRL and PGW at ski resort base from 14 January 0000 UTC, 2010 to 16 January 1200 UTC, 2010. Precipitation occurrence is shown as rain (blue), green (snow), red (transition region) and grey (no precipitation). Ski resorts are separated into western sub-area (top) and eastern sub-area (bottom) and within each sub-area are organized from west (top) to east (bottom).

Figure with precipitation accumulation data showing:

Legend:
- Rain - WRF CTRL
- Melted Graupel Equivalent - WRF CTRL
- Melted Snow Equivalent - WRF CTRL
- Total precipitation - ECCC observation

Y-axis: Precipitation Accumulation (mm), ranging 0 to 1400

X-axis categories: Cypress Bowl North, Whistler Mid Station, Revelstoke, Jasper, Glacier NP Rogers Pass*, Golden*, Yoho NP, Banff, Fernie*

Elevation labels: 729 m, 953 m, 1133 m, 1320 m, 592 m, 445 m, 1860 m, 1330 m, 1226 m, 922 m, 785 m, 1736 m, 1602 m, 1020 m, 1397 m, 1580 m, 1199 m, 1001 m

**Fig. 1.** Figure 4. Total precipitation accumulation from January to April, 2010. The left-hand bars represent the closest CTRL grid point, separated into three components, including rain ...

[Figure]

**Fig. 2.** Figure 10. Transition region occurrences for the (a) western and (b) eastern sub-areas binned according to elevation. Vertical solid blue lines and dashed red lines represent the average elevation ...

[Figure]

**Fig. 3.** Figure 14: Timeline of precipitation type under CTRL and PGW at ski resort base from 14 January 0000 UTC, 2010 to 16 January 1200 UTC, 2010. Precipitation occurrence is shown as rain (blue), green ...